# A genome-wide association study identifies the *GPM6A* locus associated with age at onset in ALS
Ryoichi Nakamura [1,32], Genki Tohnai[2,32], Naoki Atsuta[1,32], Yumi Matsuda [3,32], Satoru Morimoto[4,5], Daisuke Ito[6], Masahisa Katsuno [6,7], Yuishin Izumi[8], Mitsuya Morita[9], Ikuko Iwata[10], Ichiro Yabe [10], Tomoko Nakazato[11], Nobutaka Hattori [11], Takehisa Hirayama[12], Osamu Kano[12], Asako Tamura[13], Naoki Suzuki [14,15], Masashi Aoki[14], Kazumoto Shibuya[16], Satoshi Kuwabara[16], Masaya Oda[17], Rina Hashimoto[18], Ikuko Aiba[18], Tomohiko Ishihara[19,20], Osamu Onodera [19], Toru Yamashita [21], Hiroyuki Ishiura[21], Kota Bokuda[22], Toshio Shimizu[22], Yoshio Ikeda[23], Kazuko Hasegawa[24], Fumiaki Tanaka[25], Takanori Yokota[26], Kazuaki Kanai[27], Yu-ichi Noto[28], Ryuji Kaji [8], Hirohisa Watanabe[29], Tomoko Konishi[2], Mikiko Hasegawa[2], Hozuki Fukaya[2], Jun-ichi Niwa [1], Manabu Doyu[1], Yohei Okada [1,30], Shiho Nakamura[4,5], Fumiko Ozawa[4,5], Hideyuki Okano [4,5], Masahiro Nakatochi [3,33] ✉ & Gen Sobue [2,31,33] ✉ On behalf of the Japanese Consortium for Amyotrophic Lateral Sclerosis research (JaCALS) study group*

Amyotrophic lateral sclerosis (ALS) exhibits considerable clinical variability, such as differences in age at onset (AAO). Multiple factors, including genetic factors, may underlie this variability; however, the specific determinants remain unclear. To identify genes affecting AAO, we have conducted a genome-wide association study in Japanese patients with ALS (discovery cohort: n = 1808; replication cohort: n = 207). Here, we show that the minor A allele of rs113161727 at the *ADAM29-GPM6A* locus is associated with a younger AAO in the discovery cohort (effect, -4.27 years; p = 4.60 × 10$^{-8}$); this finding has been confirmed in the replication cohort (p = 0.0068) and meta-analysis (p = 1.08 × 10$^{-9}$). Among 65 ALS patients with a *SOD1* mutation, the AAO has been found to be 10.2 years younger in those with the A allele than in those without it (p = 0.002). This variant correlates with *GPM6A* upregulation in iPSC-derived motor neurons, suggesting *GPM6A* as a candidate AAO modifier. Overall, our study highlights the impact of genetic modifiers on ALS heterogeneity and provides a potential target for delaying disease onset.

Amyotrophic lateral sclerosis (ALS) is a devastating neurodegenerative disorder characterized by the progressive degeneration of motor neurons. This degeneration leads to symptoms including muscle weakness, bulbar palsy, and ultimately, death due to respiratory failure[1]. The clinical features of ALS are heterogeneous in terms of age at onset (AAO), site of onset, progression patterns, and survival time. Additionally, the AAO of ALS has been reported to affect various clinical features. The AAO is a common prognostic factor for both functional decline and survival in patients with ALS[2]. An older AAO of ALS is associated with a higher rate of bulbar onset and a faster decline in bulbar function[3,4].

In Japanese patients with ALS, the AAO varies significantly, ranging from 20 to 80 years, with a median AAO of 62.1 years[3]. This may be influenced by various factors, including genetics. Several studies have reported that patients with ALS harboring multiple rare variants of ALS-associated genes have younger AAOs, indicating a potential cumulative effect[5,6]. Mutations in ALS-causative genes, such as *SOD1* and *FUS*, have been shown to affect AAO[7–9]. However, considerable variation in AAO has been reported even among patients with an identical mutation in those genes[7]. These findings suggest the involvement of other genetic factors in modulating the AAO in patients with ALS, although these factors remain largely unidentified.

A full list of affiliations appears at the end of the paper. *A list of authors and their affiliations appears at the end of the paper.
✉e-mail: mnakatochi@met.nagoya-u.ac.jp; sobueg@aichi-med-u.ac.jp

Genome-wide association studies (GWASs) have emerged as powerful tools for identifying genetic variants associated with disease traits[10,11]. These studies have also been used to search for modifier genes that affect ALS phenotypes, such as progression, survival, and AAO[12–14]. A recent GWAS conducted in European ALS cohorts identified genetic variants in *CTIF* as factors associated with AAO[14]. Given that the genetic factors influencing ALS risk and clinical presentation differ among populations[12,15], findings from genetic studies in European populations are not always fully applicable to Asian populations. Therefore, a GWAS in Japanese patients with ALS is important to elucidate population-specific or population-common genetic factors.

In this study, we aimed to identify the modifier genes that affect AAO in Japanese patients with ALS by conducting a GWAS. We identified rs113161727 at the *ADAM29-GPM6A* locus, which is associated with earlier AAO in this ALS population, and validated this association in an independent Japanese cohort. This variant was associated with the upregulated expression of *GPM6A* in induced pluripotent stem cell (iPSC)-derived motor neurons from patients. These findings enhance our understanding of the genetic modifiers underlying ALS and suggest promising directions for the development of targeted therapeutic strategies.

## Results
### Baseline characteristics
This study ultimately included 1808 and 207 Japanese patients with ALS in the discovery and replication cohorts, respectively. The mean AAO (years ±standard deviation (SD)) was 62.0 ± 12.4 years and 64.4 ± 11.3 years, with a sex ratio (male/female) of 1.39 and 1.25 in the discovery and replication cohorts, respectively. Generally, the AAO was younger in males than in females (61.5 vs. 63.2 years, $p = 0.002$). The baseline characteristics of the study participants are summarized in Supplementary Table 1.

We included 65 patients with ALS and *SOD1* mutations (*SOD1*-ALS) in the discovery cohort. Their mean AAO (55.4 ± 12.0 years) was significantly younger than that of patients without *SOD1* mutations (62.2 ± 12.4 years in the discovery cohort, $p = 1.36 \times 10^{-5}$; 64.4 ± 11.3 years in the replication cohort, $p = 8.30 \times 10^{-8}$; Student's *t*-test; Supplementary Table 2). Detailed information on *SOD1* mutations is provided in Supplementary Table 3.

### GWAS for AAO in patients with ALS
We performed a GWAS for the AAO of ALS in the discovery and replication cohorts. Subsequently, the results from these cohorts were pooled in a meta-analysis. In the discovery phase, a single locus at 4q34.2 achieved genome-wide significance with a *p*-value < $5 \times 10^{-8}$ for the AAO of ALS (Fig. 1a and Supplementary Fig. 1). The lead single-nucleotide polymorphism (SNP) with the lowest *p*-value for the AAO of ALS was rs113161727 ($p = 4.60 \times 10^{-8}$). The minor allele A was associated with a younger AAO, which was found to reduce the AAO by 4.27 years (standard error (SE) = 0.78; Table 1, Fig. 2c, and Supplementary Fig. 2a).

In the replication phase, this association was validated in an independent cohort of 207 Japanese patients with ALS (Table 1, $p = 6.81 \times 10^{-3}$, $\beta = -5.10$, SE = 1.87). A meta-analysis combining both cohorts further confirmed the genome-wide significance of the SNPs at the 4q34.2 locus, with the lead SNP being rs113161727 (Fig. 1b, c and Table 1; $p = 1.08 \times 10^{-9}$, $\beta = -4.40$, SE = 0.72). The quantile–quantile plot for the *p*-values is shown in Supplementary Fig. 3. All 18 SNPs that achieved genome-wide significance (Supplementary Data 1) were located in the intergenic region between *ADAM29* and *GPM6A* (Fig. 1c).

Fig. 2a shows the distribution of AAO. The peak AAO was within the 65–70-year age range. Fig. 2b shows the distribution of AAO divided by the rs113161727 genotype. The distribution of the AAO of patients with ALS with the GG genotype of rs113161727 (Fig. 2b) was almost identical to that of the AAO in all patients (Fig. 2a, Supplementary Data 2). Conversely, patients with the AG or AA genotype exhibited a younger AAO distribution than those with the GG genotype. An AAO < 40 years was 11.8% in patients with the AG or AA genotype compared to 5.3% in those with the GG

genotype. In contrast, an AAO ≥ 70 years was 17.7% in patients with the AG or AA genotype compared to 29.7% in those with the GG genotype (Supplementary Data 2).

Given the variability in AAO among patients with *SOD1*-ALS, we conducted a subgroup analysis to assess the association between rs113161727 and AAO. The AAO of *SOD1*-ALS patients with the AA or AG genotype of rs113161727 was 10.2 years younger than those with the GG genotype ($p = 0.002$; Supplementary Table 4). Box plots and cumulative incidence curves are shown in Fig. 2d and Supplementary Fig. 2b. We also conducted a meta-analysis excluding patients with *SOD1*-ALS. The meta-analysis confirmed genome-wide significance at the 4q34.2 locus (rs113161727, $p = 2.23 \times 10^{-8}$, $\beta = -4.11$, SE = 0.73; Supplementary Fig. 4, Supplementary Table 5).

Additionally, we examined previously reported candidate SNPs associated with AAO in patients with ALS (Supplementary Table 6). The previously reported SNPs in European and Chinese patients with ALS, such as rs2046243 in *CTIF* and rs10128627 in *FRMD8*, showed no significant association with AAO in our study (rs2046243: $p = 0.466$, effect = $-0.393$, SE = 0.538; rs10128627: $p = 0.384$, effect = 0.597, SE = 0.686).

### Association between rs113161727 and ALS phenotypes
Since AAO affects various clinical phenotypes of ALS, including the onset site and survival time, we further investigated the impact of rs113161727 on the clinical phenotypes of ALS. Fig. 3 shows a forest plot of the association between rs113161727 and the site of onset (Supplementary Data 3). The minor allele A of rs113161727 significantly increased lower limb onset ($p = 0.005$) and tended to decrease bulbar onset ($p = 0.048$); however, it did not affect upper limb onset. Additionally, rs113161727 did not significantly affect survival time from onset, with a hazard ratio of 1.21 (95% confidence interval, 0.95–1.32, $p = 0.167$).

### Gene expression analysis of induced pluripotent stem cell-derived motor neurons
We investigated the effects of rs113161727 on the expression of its surrounding genes (*GLRA3*, *ADAM29*, *GPM6A*, *WDR17*, and *SPATA4*) in iPSC-derived motor neurons of patients with ALS. Gene expression levels were assessed by real-time quantitative reverse transcription PCR (RT-qPCR). We established iPSCs from lymphoblastoid B cell lines (LCLs) derived from 20 patients with ALS following established protocols[16,17]. Subsequently, we derived motor neurons from these iPSCs. The rs113161727 genotypes in both LCLs and iPSC-derived motor neurons were confirmed by Sanger sequencing (Supplementary Fig. 5). Ten patients carried the AG genotype, while the other ten had the GG genotype. In iPSC-derived motor neurons, the expression levels of *GPM6A* were significantly higher in the presence of the rs113161727 AG genotype ($N = 10$) than in that of the GG genotype ($N = 10$) (Fig. 4 and Supplementary Data 4; $p = 0.0039$, Mann–Whitney *U*-test). In contrast, rs113161727 did not affect the expression of other surrounding genes (*GLRA3*, *ADAM29*, *WDR17*, and *SPATA4*). RT-qPCR analyses for all genes were performed with three technical replicates per sample. These results suggest that rs113161727 is associated with the upregulated expression of *GPM6A*.

To validate this finding, we further analyzed RNA sequencing (RNA-seq) data for iPSC-derived motor neurons from 67 Japanese patients with ALS. Consistent with our RT-qPCR results, the RNA-seq analysis confirmed that *GPM6A* expression was significantly higher in patients with the AG genotype ($N = 9$) than in those with the GG genotype ($N = 58$) ($p = 0.029$, Mann–Whitney *U*-test, Supplementary Fig. 6, Supplementary Data 5). At the protein level, western blotting of iPSC-derived motor neurons showed a similar trend without reaching statistical significance (AG: $N = 5$; GG: $N = 5$; Mann–Whitney *U*-test, $p = 0.22$; Supplementary Fig. 7).

## Discussion
In this study, we identified rs113161727, a genetic variant associated with AAO in Japanese patients with ALS. This variant was located in the

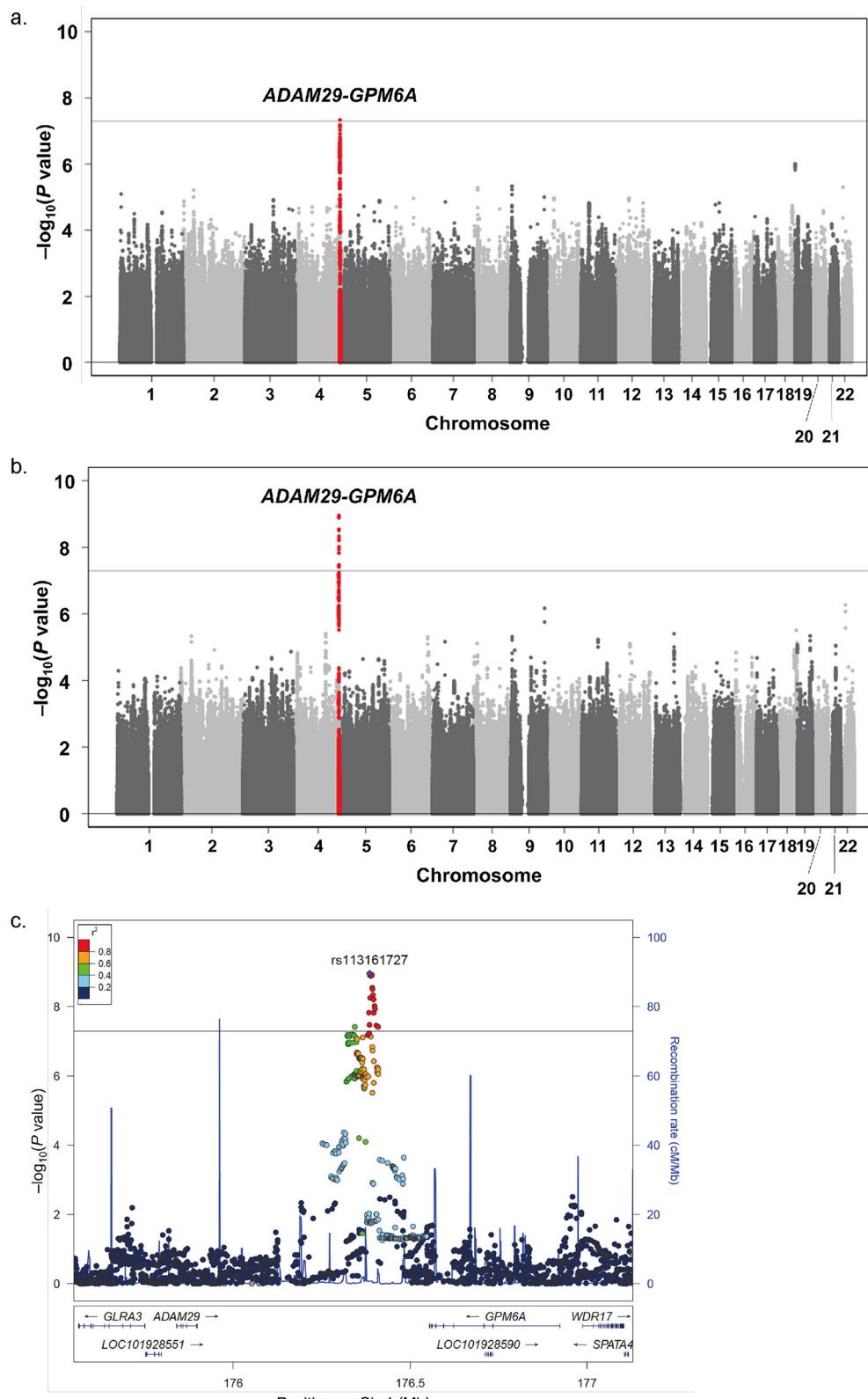

**Fig. 1 | Genome-wide association analysis for age at onset in Japanese patients with amyotrophic lateral sclerosis (ALS). a** Manhattan plot for age at onset in patients with ALS in the discovery cohort ($n = 1808$). The horizontal gray line represents the genome-wide significance level ($\alpha = 5 \times 10^{-8}$). Loci significantly associated with the age at ALS onset are highlighted in red. One locus (4q34.2) showed a genome-wide significance. **b** Manhattan plot of the meta-analysis of the age at onset in Japanese patients with ALS. We combined the results from the discovery ($n = 1808$) and replication ($n = 207$) cohorts in a meta-analysis (total

$n = 2015$ independent patients). The meta-analysis confirmed the genome-wide significance of the single-nucleotide polymorphisms (SNPs) at the 4q34.2 locus, with the leading SNP being rs113161727. **c** Regional association plots for the 4q34.2 locus identified in the genome-wide meta-analysis ($n = 2015$). The vertical axis represents $-\log_{10}$ ($p$-value) for assessing the association between each SNP and age at onset. Colors indicate the linkage disequilibrium ($r^2$) between each sentinel SNP and neighboring SNPs based on the JPT population of the 1000 Genomes Project phase 3. JPT Japanese people in Tokyo, Japan.

**Table 1 | Genomic region and lead SNP for the age at onset of patients with ALS**

| SNP | Chr | Position | Gene | Alleles | | Group | EAF | Effect | SE | *p*-value | *I²* | HetPVal |
|-----|-----|----------|------|---------|---|-------|-----|--------|-----|-----------|-----|---------|
| | | | | Non-effect | Effect | | | | | | | |
| rs113161727 | 4 | 176385733 | *ADAM29, GPM6A* | G | A | Discovery cohort | 0.084 | −4.271 | 0.782 | $4.60 \times 10^{-8}$ | | |
| | | | | | | Replication cohort | 0.085 | −5.104 | 1.867 | $6.81 \times 10^{-3}$ | | |
| | | | | | | Meta-analysis | 0.084 | −4.395 | 0.721 | $1.08 \times 10^{-9}$ | 0 | 0.681 |

Position is based on Human Genome Assembly build 37. Gene symbols are shown in italics.

*SNP* single-nucleotide polymorphism, *Chr* chromosome, *EAF* effect allele frequency, *SE* standard error, HetPVal *p*-value from the test of heterogeneity.

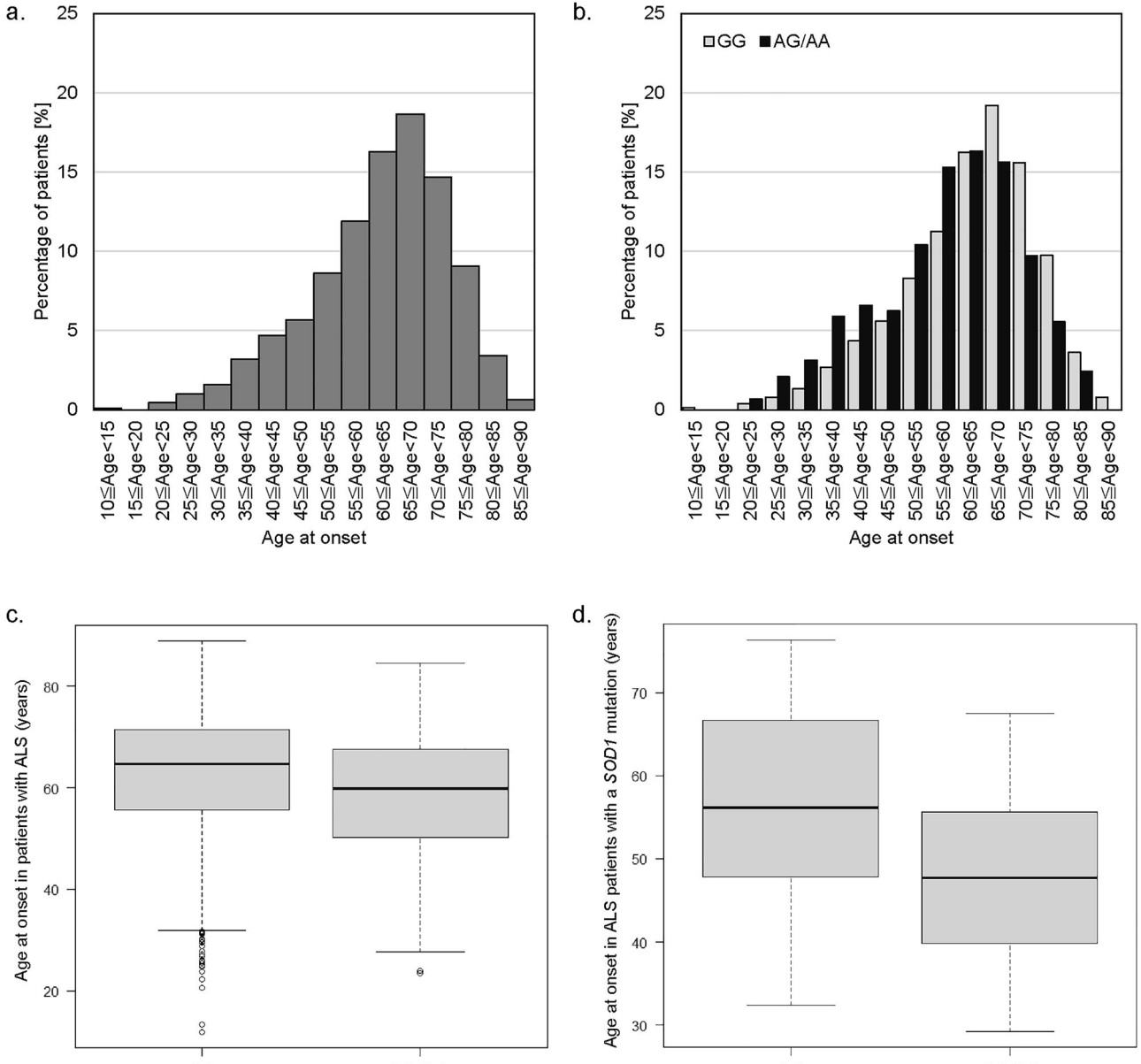

**Fig. 2 | Distribution of age at onset (AAO) and the relationship between AAO and rs113161727 in Japanese patients with amyotrophic lateral sclerosis (ALS).** **a** Distribution of AAO in the discovery cohort (*n* = 1808). **b** Distribution of AAO in the discovery cohort according to the rs113161727 genotype. Patients with the AG or AA genotype of rs113161727 showed a younger AAO distribution than those with the GG genotype. AAO < 40 years was 11.8% in patients with the AG or AA genotype, compared to 5.3% in those with the GG genotype. The AAO ≥ 70 years was 17.7% in patients with the AG or AA genotype, compared with 29.7% in those with the GG genotype. **c** Relationship between AAO and rs113161727 in the discovery cohort (*n* = 1808). **d** Relationship between AAO and rs113161727 in patients with ALS with a SOD1 mutation (*n* = 65). The bottom and top of the box indicate the interquartile ranges (25th and 75th percentiles), and the line represents the median. Whiskers under and over the box correspond to a 1.5× interquartile range, and circles indicate outliers.

intergenic region between *ADAM29* and *GPM6A*. The expression analysis suggested that rs113161727 affects the expression levels of *GPM6A*.

Several studies have reported genetic factors affecting AAO in patients with ALS[14,18–20]. Previous GWASs conducted in patients with ALS of European ancestry failed to identify any genome-wide significant loci that influenced the AAO of ALS[18,19]. However, a recent study revealed a significant association between *CTIF* polymorphisms and AAO in a large cohort comprising 9353 European patients with ALS[14]. Additionally, a GWAS in 2788 patients with ALS of Chinese ancestry identified SNPs in *FRMD8* associated with earlier AAO (by 3.15 years) for ALS[20]. However, our study did not replicate these recently reported SNPs, suggesting that the genetic factors affecting the AAO of Japanese patients with ALS are distinct from those of patients with ALS in other populations. This discrepancy underscores the importance of genetic diversity in the modifier genes that affect ALS phenotypes and highlights the potential significance of population-specific genetic factors. Previous studies have shown that some genetic modifiers associated with ALS in European and Chinese populations are not associated with ALS in Japanese cohorts. For instance, the rs12608932 variant in *UNC13A* is associated with ALS incidence and survival in European populations[21–24], but not in Japanese and Chinese populations[11,12,25]. Similarly, while intermediate repeat expansions in *ATXN2* are associated with an increased risk of ALS and shorter survival in both European and Chinese populations[26–29], no such associations have been found in Japanese patients with ALS[30].

The failure to replicate the association between previously reported SNPs and ALS could also be explained by the difference in AAO across populations. In fact, the distribution of the AAO for ALS in Japan differs from that in other countries[14,20]. Phenotypes, such as the AAO and prognosis of ALS, show considerable variation, even among Asian countries[31]. In our study, the mean AAO was 62.0 years, with the peak AAO in the 65–70-year age group in the Japanese patients with ALS. In contrast, the mean AAO was 54.6 years, with peak ages in the 50s and 60s in the Chinese patients with ALS[20], and 59.9 years, with a peak age in the 60s in the European patients with ALS[14]. As one of the most aged societies with the highest life expectancy globally[32], Japan's demographic profile may contribute to an older AAO distribution in patients with ALS and influence the detection of genetic factors.

We found that patients with *SOD1*-ALS harboring rs113161727 exhibit an approximately 10-year earlier AAO. Patients with ALS harboring multiple rare variants of ALS-associated genes can develop AAO earlier[5,6]. Intriguingly, even among individuals harboring an identical mutation in *SOD1*, the AAO of patients with ALS exhibits considerable variability[7]. The multistep hypothesis has been proposed that six distinct steps lead to the onset of ALS[33]; however, patients with *SOD1*-ALS only need two steps on average for the disease to manifest[34]. Our findings suggest that rs113161727 may act in one of the remaining two steps, thereby exacerbating disease onset in patients with *SOD1*-ALS. A recent study demonstrated the effectiveness of the antisense oligonucleotide tofersen in patients with *SOD1*-ALS[35]. Given the variability in AAO among patients with *SOD1*-ALS, our results suggest the potential need for earlier intervention in patients with *SOD1*-ALS with rs113161727. The implications of our study extend to the timing and approach of applying treatments, such as tofersen, in patients with specific genetic profiles, emphasizing the importance of personalized medicine in patients with ALS.

The lead SNP in our study, rs113161727, affected the expression of *GPM6A*. *GPM6A* encodes glycoprotein M6a (GPM6A). Several SNPs in *GPM6A* have been associated with various neuropsychiatric diseases, such as schizophrenia and depression[36,37]. Patients with ALS and their relatives often have comorbidities such as anxiety, depression, cognitive dysfunction, and suicidal ideation[38,39]. Recent studies have shown that ALS, neuropsychiatric diseases, and cognitive dysfunction share common genetic backgrounds[10,40,41]. These findings suggest that *GPM6A* may play a role in shared pathways between ALS and neuropsychiatric diseases. Further studies are needed to elucidate its role and underlying mechanisms.

Neuronal GPM6A is a member of the tetraspan proteolipid protein family, which is involved in neuronal development, synapse formation, and plasticity[42]. GPM6A is primarily distributed throughout the human central nervous system. It is found in presynaptic membranes and is enriched in glutamatergic synaptic vesicles[42,43]. Glutamate excitotoxicity is responsible for neuronal death in ALS[44]. An increase in GPM6A may heighten glutamate excitotoxicity, leading to more glutamate release, which exacerbates

| Site | N (+) | N (−) | OR (95%CI) | P-value |
|------|-------|-------|------------|---------|
| Upper limb | 747 | 911 | 0.9 (0.69-1.17) | 0.43 |
| Lower limb | 484 | 1174 | 1.47 (1.12-1.93) | 0.005 |
| Bulbar palsy | 436 | 1222 | 0.73 (0.53-1.00) | 0.048 |

**Fig. 3 | Forest plots of the association between rs113161727 and the site of onset.** Forest plot shows odds ratios (ORs) with 95% confidence intervals (CIs) for the A allele of rs113161727 in relation to the site of onset. The analyses were conducted using the discovery cohort, which included 1658 patients with available clinical data. N (+) indicates the number of patients with the onset site, and N (−) indicates the number of patients without the onset site. Black squares represent the OR, and dashed horizontal error bars indicate the 95% CI. The vertical gray line indicates OR = 1. The A allele of rs113161727 significantly increased the risk of lower limb onset (*p* = 0.005) and tended to decrease the risk of bulbar onset (*p* = 0.048) but did not affect upper limb onset. OR odds ratio, 95% CI 95% confidence interval.

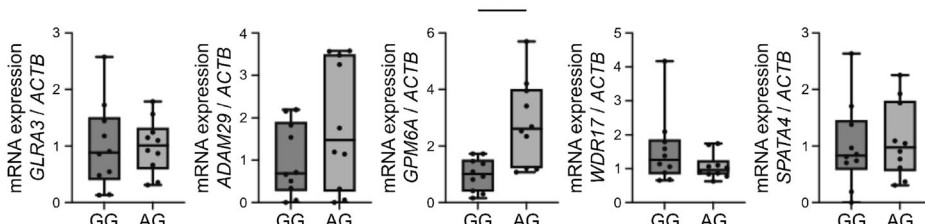

**Fig. 4 | Relative expression levels of genes surrounding rs113161727 in induced pluripotent stem cell (iPSC)-derived motor neurons from patients with amyotrophic lateral sclerosis (ALS) with each genotype of rs113161727.** The expression levels of genes (*GLRA3*, *ADAM29*, *GPM6A*, *WDR17*, and *SPATA4*) surrounding rs113161727 in iPSC-derived motor neurons from 20 patients with ALS were examined using real-time quantitative reverse transcription-polymerase chain reaction (RT-qPCR). The expression levels of *GPM6A* mRNA were significantly higher in iPSC-derived motor neurons with the AG genotype of rs113161727 (*n* = 10) than in those with the GG genotype (*n* = 10, *p* = 0.0039). In contrast, expression levels of the other four genes (*GLRA3*, *ADAM29*, *WDR17*, and *SPATA4*) were not affected by the rs113161727 genotype. The mRNA expression levels of each genotype were compared using the Mann–Whitney *U*-test. The mRNA levels of each gene were normalized to the levels of *ACTB*. RT-qPCR analysis included technical replicates conducted in three wells for each plate. The Cq values from the three technical replicates were averaged to generate a single raw Cq value per assay. Each black dot represents an individual sample. The bottom and top of the box indicate the interquartile range (25th and 75th percentiles), and the line represents the median. The whiskers under and over the box correspond to the minimum and maximum values. Asterisk (*) indicates *p* < 0.01.

motor neuron damage. The AA and AG genotypes of the identified SNP tended to confer a younger age of onset, particularly in patients with *SOD1*-ALS. Patients with *SOD1*-ALS are affected by glutamate excitotoxicity due to the inactivation of glutamate transporter 1 by *SOD1* mutations[45]. Elevated levels of extracellular glutamate in the cortex of *SOD1*-G93A transgenic mice indicate a potential role of glutamate excitotoxicity in *SOD1*-ALS[46]. Patients with ALS bearing the exacerbating genotypes of the identified SNP may have heightened glutamate hyperexcitability, leading to increased glutamate release and the exacerbation of motor neuron damage and, consequently, a younger AAO.

The strength of our research is that the GWAS was performed using data from a cohort study in a single ethnic group and was validated in another cohort from the same ethnicity. Another strength is that expression data were examined in iPSC-derived motor neurons from patients with ALS. Our recent study showed that clinical phenotypes correlate with iPSC-based models, even in patients with sporadic ALS[12,16,17]. Analysis using tissue-specific cells differentiated from iPSCs in the search for modifier genes, as in this study, will be a useful method for modifier genes of the ALS phenotype in the future.

While our findings emphasize the importance of population-specific genetic factors, our study has some limitations. The relatively modest sample size may have reduced the power to detect additional associations and contributed to the lack of replication of previously reported SNPs. Addressing these limitations will require larger, collaborative studies to identify robust genetic modifiers of ALS phenotypes such as AAO and survival.

In conclusion, we identified rs113161727 at the *ADAM29-GPM6A* locus associated with AAO in patients with ALS, which affected the expression of *GPM6A* in iPSC-derived motor neurons. Our study highlights the impact of genetic modifiers on ALS phenotypes, such as AAO, and provides a potential target for developing treatments that delay the onset of ALS.

## Methods
### Patients
The GWAS discovery phase included 2087 Japanese patients with ALS who were enrolled in the Japanese Consortium for Amyotrophic Lateral Sclerosis Research (JaCALS) registry. For the replication phase, 222 patients from another cohort derived from two Japanese institutions (Jichi Medical University and Hokkaido University) were included. The JaCALS registry is a multicenter patient registry for ALS involving 42 institutions across Japan. Patients with ALS contribute DNA samples and LCLs, and they are contacted every 3 months for a telephone survey. All patients were diagnosed according to the revised El Escorial diagnostic criteria[47]. AAO was defined as the age at which the patients became initially aware of muscle weakness or impairment of swallowing, speech, or respiration[3,48]. Genomic DNA was extracted from peripheral blood leukocytes, following standard protocols[49].

All participants provided written informed consent. This study protocol received ethics approval from the Ethics Review Committee of Aichi Medical University School of Medicine (approval number: 2021-083) and from the ethics committees of all participating institutions. All ethical regulations relevant to human research participants were followed.

### Genotyping, quality control, and whole-genome imputation
Overall, 2087 and 222 patients from the discovery and replication cohorts, respectively, underwent genotyping at the RIKEN Institute (Yokohama, Japan) and RIKEN GENESIS (Tokyo, Japan). The genotyping was performed using a HumanOmniExpressExome BeadChip array (Illumina, San Diego, CA, USA). One sample was excluded because it had a genotype call rate of <0.98. Nine samples exhibited discordance between genetically inferred and self-reported sex. Using the identity-by-descent method in PLINK 1.9 software[50], 32 duplicate or closely related pairs of samples were detected (PI_HAT > 0.1875), and one sample of each pair was excluded. Principal component analysis referencing the 1000 Genomes Project reference phase 3 panel[51] indicated 13 patients whose inferred ancestry fell outside the Japanese population. These patients were subsequently excluded. Non-autosomal SNPs, SNPs with a genotype call rate <0.98 or Hardy–Weinberg equilibrium exact test $p$-value < $1 \times 10^{-6}$, minor allele frequency (MAF) < 0.01, or a deviation from the allele frequency computed from the 1000 Genomes Project phase 3 East Asians (EAS) samples were excluded. Quality control filtering resulted in the selection of 2039 patients from the discovery cohort, 216 patients from the replication cohort, and 522,030 SNPs. Genotype imputation was performed using SHAPEIT2[52] and Minimac3[53] software, employing the cosmopolitan reference panel from the 1000 Genomes Project (phase 3)[51]. Variants with $r^2 < 0.8$ or MAF < 0.01 were excluded. This genotype imputation yielded 6,963,364 and 6,916,778 variants in the discovery and replication cohorts, respectively. Overall, 2015 patients with ALS (1808 and 207 from the discovery and replication cohorts, respectively) were included in the AAO GWAS. All patients included in the study were of Japanese ancestry with ALS. ALS diagnoses were categorized according to the revised El Escorial diagnostic criteria[47] as definite, probable, probable laboratory-supported, or possible. In the discovery and replication cohorts, 65 of 1808 patients and 1 of 207 patients harbored a *SOD1* mutation, respectively. The details of the *SOD1* mutation are provided in Supplementary Table 3.

### GWAS for AAO of ALS
The association between AAO and SNPs was assessed using the BOLT-LMM algorithm[54] in patients with ALS from the discovery cohort. Association analysis was performed using linear regression analysis with the Efficient and Parallelizable Association Container Toolbox (EPACTS) (https://genome.sph.umich.edu/wiki/EPACTS) for patients with ALS from the replication cohort. The independent variables included the imputed dosage genotypes of the SNPs and covariates. Specifically, the covariates of the model included sex and the first two principal components derived from the genotyped data for the discovery cohort, and the first ten principal components for the replication cohort. A fixed-effects meta-analysis of the association between AAO and SNPs was conducted utilizing METAL[55], employing the inverse variance-weighted approach. The presence of heterogeneity was evaluated using Cochran's Q-test. Ultimately, a meta-analysis included 6,843,603 SNPs from both cohorts. We considered variants with $p$-value < $5 \times 10^{-8}$ as genome-wide significant.

We employed various bioinformatic approaches to collate functional annotations and rank the associated SNPs at a novel locus. We used ANNOVAR[56] to compile a comprehensive set of functional annotations, including the locations of lead SNPs.

### Association analysis of SNPs with other phenotypes
The association between the site of onset and SNPs was assessed using logistic regression with PLINK2[50] in patients with ALS from the discovery cohort.

Subsequently, we estimated the effect of the lead SNP with genome-wide significance on survival using the Kaplan–Meier method with the log-rank test. The association between tracheostomy and invasive ventilation-free survival and SNPs was investigated using Cox proportional hazards regression analysis in *R* with the survival package.

We analyzed the association between SNPs and survival time as well as the three sites of onset—the upper limb, lower limb, and bulbar onset. After applying the Bonferroni correction for multiple comparisons, statistical significance was determined at a threshold of $p < 0.0125$ ( = 0.05/4).

### Preparation of LCLs from patients with ALS
Peripheral blood mononuclear cells (PBMCs) were obtained from patients with ALS. Patient-derived PBMCs were immortalized via Epstein–Barr virus infection according to the protocol of SRL, Inc. (Tokyo, Japan) and transformed into LCLs as described below.

PBMCs were washed with RPMI 1640 (Thermo Fisher Scientific, Waltham, MA, USA) and resuspended at $1 \times 10^6$ cells/mL in RPMI

1640 supplemented with 20% fetal calf serum (FCS, Nichirei Biosciences, Tokyo, Japan) and 200 ng/mL cyclosporin A (CsA, Novartis Pharmaceuticals, Basel, Switzerland) (pH 6.4). PBMCs were infected with Epstein–Barr virus (medium:virus = 4:1) and incubated at 37 °C. After 3–4 days, 0.5 mL of CsA-supplemented medium was added. From day 7, half of the medium was replaced twice weekly. Small cell clumps appeared within 3–7 days, and by 2–3 weeks, larger aggregates became visible. The medium was then switched to RPMI 1640 supplemented with 10% FBS. Once aggregates were visible, cultures were expanded in 25 cm² flasks and maintained at 3–5 × 10⁵ cells/mL with biweekly medium changes. LCLs were cryopreserved in RPMI 1640 with 10% FBS and 10% DMSO (Sigma-Aldrich, St Louis, MO, USA).

### Establishment and culture of iPSCs from human LCLs

iPSCs were generated from LCLs according to the protocol reported by Morimoto et al.[16], using the steps below. LCLs were cultured in RPMI 1640 medium (Thermo Fisher Scientific) supplemented with 10% FBS at 37 °C and 5% $CO_2$ in a humidified incubator. After several passages, the LCLs were electroporated with 0.63 mg each of pCE-hOCT3/4, pCE-hSK, pCE-hUL, pCEmp53DD, or 0.5 mg pCXB-EBNA1 (Addgene #41813, 41814, 41855, 41856, 41857) using an Amaxa Human B cell Nucleofector Kit and Nucleofector 2b device (Lonza, Basel, Switzerland). Transfection was performed according to the manufacturer's protocol.

Transfected cells were seeded on an iMatrix-511 silk (Laminin511E8; Matrixome, Osaka, Japan)-coated plate in KBM502 with the beads. StemFit AK02N medium (Ajinomoto, Tokyo, Japan) was added on days 3, 5, and 7. The medium was fully changed every other day from day 9 onward. From the emergence of iPSC colonies at approximately 20–30 days after transfection, each cell was mechanically isolated and passaged every 7 days.

### Motor neuron differentiation for real-time quantitative reverse transcription-polymerase chain reaction

Motor neurons were generated according to the protocol reported by Sato et al.[57], using the following steps with minor modifications described herein. iPSCs were seeded in iMatrix-511-coated 12-well plates (IWAI North America Inc., Burlingame, CA, USA) at a density of $1 \times 10^5$ cells/well in the StemFit AK02N medium (Ajinomoto Co., Inc., Chuo, Japan). After 3 days, neural induction was initiated by changing the medium to neural induction medium [comprising Advanced Dulbecco's Modified Eagle Medium (DMEM)/F-12 (Thermo Fisher Scientific), 2% B27 supplement (–vitamin A) (Thermo Fisher Scientific)] with 150 nM LDN193189 (bone morphogenetic protein 2 receptor (ALK2/3)) inhibitor (StemRD, Burlingame, CA, USA), and 5 µM SB431542 (transforming growth factor-β receptor (ALK4/5/7) inhibitor; Tocris, Bristol, UK). Day 0 was defined as the initiation of neural induction culture. The medium included: 3 µM CHIR99021 (glycogen synthase kinase 3β inhibitor activating Wnt signaling; Stemgent, Cambridge, MA, USA), 1 µM Retinoic acid (Sigma-Aldrich) from days 0 to 12, and 1 µM purmorphamine (Smoothened agonist activating Shh signaling; Calbiochem, San Diego, CA, USA) from days 2 to 12. On day 6, Accutase (Nacalai Tesque, Inc., Kyoto, Japan) was used to generate single cells, which were subsequently seeded at 1:2 onto 12-well plates coated with poly-L-ornithine and laminin. Following a repeat dissociation on day 12, $2 \times 10^5$ cells/well were replated onto poly-l-ornithine- and laminin-coated 48-well plates and maintained in neuronal medium (advanced DMEM/F-12, 2% B27 supplement, 200 µM ascorbic acid (Sigma-Aldrich), and 200 µM dbcAMP (Sigma-Aldrich)) with 20 µM N-[N-(3,5-difluorophenacetyl)-l-alanyl]-S-phenylglycine t-butyl ester (DAPT, γ-secretase inhibitor; Sigma-Aldrich). DAPT was removed on day 18. Motor neuron differentiation was performed once per iPSC line, using 20 independent patient-derived iPSC lines (AG: $n = 10$; GG: $n = 10$).

### Validation of LCL and iPSC-derived motor neuron genotypes

iPSC-derived motor neurons were established from LCLs derived from 20 Japanese patients with ALS. The rs113161727 genotypes in both LCLs and iPSC-derived motor neurons were verified by PCR amplification followed by Sanger sequencing. The target region was amplified using the forward

primer 5′- CAAGCCAAGCAGAGGAGTGT -3′ and reverse primer 5′-CTGAGTGACCCCAGCTTTCA-3′, and the amplicons were sequenced with the primer 5′-AGCCCAGGATTCCTCCTAAG-3′.

### Real-time quantitative reverse transcription-polymerase chain reaction

We analyzed the impact of the genome-wide significant SNP on the expression of genes surrounding the SNP (*GLRA3*, *ADAM29*, *GPM6A*, *WDR17*, and *SPATA4*) in iPSC-derived motor neurons from patients with ALS using real-time quantitative reverse transcription-polymerase chain reaction (RT-qPCR). RNA was extracted from iPSC-derived motor neurons on day 28 after differentiation induction, using the RNeasy mini kit (QIAGEN, Hilden, Germany). The extracted RNA was synthesized into cDNA using SuperScript 4 VILO (Thermo Fisher Scientific). A quantitative polymerase chain reaction was performed using the THUNDERBIRD SYBR qPCR Mix (Toyobo, Japan) and a Light Cycler 96 thermocycler (Roche, Basel, Switzerland) according to the manufacturer's instructions. The primer sequences are listed in Supplementary Table 7. *ACTB* mRNA levels were used as internal control genes. The expression levels of the genes in iPSC-derived motor neurons from patients with ALS carrying the risk allele of rs113161727 are shown as relative fold-change compared to those without the risk allele. RT-qPCR analysis included technical replicates conducted in three wells for each plate. The Cq values from the three technical replicates were averaged to generate a single raw Cq value per assay. Furthermore, each experiment was independently repeated three times to ensure reproducibility. The final value represents the mean of the three independent experiments.

We compared mRNA expression in iPSC-derived motor neurons from patients with ALS for each SNP genotype using the Mann–Whitney *U*-test. Statistical analyses were performed using the Prism 10 software (GraphPad Software, San Diego, CA, USA). We analyzed five genes surrounding the genome-wide significant SNP. Bonferroni correction for multiple comparisons was applied, and a threshold of $p < 0.01$ (=0.05/5) was considered statistically significant.

### Western blotting

Motor neurons at day 28 of differentiation were lysed in ice-cold RIPA buffer (Wako, Osaka, Japan) supplemented with cOmplete protease inhibitor cocktail (Roche) and PhosSTOP phosphatase inhibitor cocktail (Roche). Protein concentrations were determined using the DC Protein Assay Kit (Bio-Rad, Hercules, CA, USA) according to the manufacturer's instructions. Equal amounts of protein (10 µg per sample) were loaded onto 5–20% SuperSep Ace gels (Wako) and separated using sodium dodecyl sulfate–polyacrylamide gel electrophoresis. Proteins were transferred to Immobilon polyvinylidene difluoride membranes (Merck Millipore, Burlington, MA, USA) and blocked in Blocking One buffer (Nacalai Tesque) for 1 h at room temperature. Primary antibodies were anti-GPM6A (Thermo Fisher Scientific, # 720252, 1:500) and anti-β-actin (Cell Signaling Technology, # 4970, 1:1250). Secondary antibody was HRP-Conjugated anti-rabbit IgG (Cytiva, # NA9340, 1:5000). Details of all primary and secondary antibodies used are listed in Supplementary Table 8. Signals were detected with ECL Prime (GE Healthcare, Milwaukee, WI, USA) and imaged using a LuminoGraph imager (ATTO Corporation, Tokyo, Japan). Image processing and quantification were performed with CS Analyzer software (ATTO Corporation), and GPM6A expression levels were normalized to β-actin.

### Motor neuron differentiation for RNA sequencing

To validate RT-qPCR findings, we additionally performed RNA sequencing using iPSC-derived motor neurons from 67 Japanese ALS patients. While the iPSCs used for both RT-qPCR and RNA-seq experiments were generated from patient-derived LCLs using the same method, the motor neuron differentiation protocols differed between the two analyses. Motor neurons were generated according to the protocol reported by Morimoto et al.[16], using the following steps.

https://doi.org/10.1038/s42003-025-09168-4 **Article**

iPSCs were first induced into a chemically transitional embryoid-body-like state (CTraS) by treatment with SB431542 (Sigma-Aldrich, MO, USA), dorsomorphin (Sigma-Aldrich), and CHIR99021 (Focus Biomolecules, PA, USA). Following this, the cells were dissociated into single-cell suspensions using trypsin and plated onto poly-L-lysine (Sigma-Aldrich) and Matrigel (Corning, NY, USA)-coated dishes in motor neuron medium supplemented with Y-27632 (Nacalai Tesque, Kyoto, Japan). On day 0, the iPSCs were transduced with Sendai virus vectors encoding transcription factors Lhx3, Ngn2, and Islet1, or Lhx3, Ngn2, Islet1, and EGFP (multiplicity of infection = 5; ID Pharma, Tokyo, Japan). On the following day (day 1), the medium was replaced with motor neuron medium without Y-27632. On day 14, total RNA was isolated from the cell lysates using the RNeasy Micro Kit (Qiagen) with on-column DNase I treatment.

### RNA sequencing and analysis

The RNA concentration and quality were assessed using the Agilent TapeStation system (Agilent Technologies, CA, USA). RNA samples were then subjected to amplification with the SMART-Seq v4 Ultra Low Input RNA Kit (Takara Bio, Shiga, Japan). Following amplification, DNA libraries were prepared using the Nextera XT DNA Library Prep Kit (Illumina, CA, USA), and sequencing was conducted on an Illumina NovaSeq platform. All library preparation and sequencing procedures were performed by Takara Bio. Raw RNA-seq reads in FASTQ format were first quality-checked, and adapter sequences were trimmed by Trimmomatic v0.39[58]. The trimmed reads were then aligned to the human reference genome (GRCh38) using the STAR aligner v2.7.11b[59] with default parameters. Gene and transcript expression levels were quantified using RSEM v1.3.1[60], which estimates transcript abundance in Transcripts Per Million (TPM)[61]. Human gtf annotation on Ensembl v113 was obtained from Ensembl and used. TPM values for *GPM6A* (ENSG00000150625) were extracted, and differences between rs113161727 genotypes were assessed using the Mann–Whitney *U*-test.

### Statistics and reproducibility

Details of the statistical tests used in the study are provided in the respective "Methods" sections. Unless otherwise specified, all statistical tests were two-sided with α = 0.05, and multiple comparisons were adjusted as detailed in the "Methods" section. For GWAS, genome-wide significance was defined as $p < 5 \times 10^{-8}$.

RT-qPCR was performed using iPSC-derived motor neurons from 20 independent patients (AG: n = 10; GG: n = 10), with three technical replicates per assay and three independent experiments; reported values are means of independent experiments.

Software used for the data analysis of this study are as follows: PLINK version 1.90b5.1 (https://www.cog-genomics.org/plink2), SHAPEIT2 (https://speciationgenomics.github.io/phasing/), Minimac3 (https://genome.sph.umich.edu/wiki/Minimac3), BOLT-LMM version 2.4.1 (https://alkesgroup.broadinstitute.org/BOLT-LMM/BOLT-LMM_manual.html), EPACTS, LocusZoom version 1.4 (https://genome.sph.umich.edu/wiki/LocusZoom_Standalone), METAL version 2011-03-25 (http://csg.sph.umich.edu/abecasis/metal/index.html), ANNOVAR, Trimmomatic v0.39 (https://github.com/usadellab/Trimmomatic/releases), STAR v2.7.11b (https://github.com/alexdobin/STAR) and RSEM v1.3.1 (https://deweylab.github.io/RSEM/).

Box plots were created using EZR version 1.68 (Saitama Medical Center, Jichi Medical University, Saitama, Japan). Cumulative incidence curves in Supplementary Fig. 2 were generated using IBM SPSS Statistics version 29.0 (IBM Corp., Armonk, NY, USA). Statistical analyses for RT-qPCR were conducted using Prism 10 (GraphPad Software, San Diego, CA, USA).

### Ethical approval

The Ethics Review Committee of Aichi Medical University School of Medicine (approval number: 2021-083) approved this study. Additionally, the ethics committees of all participating institutions approved this study.

### Reporting summary

Further information on research design is available in the Nature Portfolio Reporting Summary linked to this article.

## Data availability

The summary statistics of our genome-wide association studies and the RNA-seq gene-level raw count matrix are available in the Human Genetic Variation Database through accession ID: HGV0000025. The source data of Figs. 2–4 have been provided as Supplementary Data 2–4. Other relevant data are available from G.S. (sobueg@aichi-med-u.ac.jp) upon reasonable request.

## Code availability

No previously unreported custom computer code or mathematical algorithm was used to generate results central to the conclusions of this study.

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

## Acknowledgements

The authors sincerely thank all patients with ALS who participated in this study, as well as the doctors and staff of JaCALS, for their invaluable assistance. Further details are listed in Supplementary Note 1. This study was supported by the Japan Agency for Medical Research and Development (AMED; grant numbers JP19km0405216, JP23ek0109492, JP23ak0101111, JP23ak0101124, JP24wm0425009, JP23ek0109538, JP24bm1123046, JP24kk0305024, JP24bm1423002, JP24bm1423003, JP24ak0101216, JP24ak0101222, JP25wn0625519, JP25ak0101216, JP25ak0101222, JP25ek0109617); Grants-in-Aid for Scientific Research KAKENHI from the Japan Society for the Promotion of Science (JSPS; grant numbers: 16H06277, 19K06523, 19K07973, 21H05278, 22K07359, 22K07509, 22H02988, 22H04923, 22H03350, 23K06835, 23K06975, 23K24249, and 25K10781); Health and Labour Sciences Research Grants from the Ministry of Health, Labour, and Welfare of Japan (grant number 23FC0201); the Hori Sciences and Arts Foundation; and the SERIKA FUND.

## Author contributions

R.N., G.T., N.A., Y.M., M.N., and G.S. conceived the study and interpreted the results. R.N., G.T., N.A., Y.M., S.M., and M.N. performed statistical and bioinformatic analyses. R.N., N.A., D.I., M.K., Yu.I., M.M., I.I., I.Y., T.N., N.H., T.H., O.K., A.T., N.S., M.A., K.S., S.K., M.O., R.H., I.A., T.I., O.O., To.Y., H.I., K.B., T.S., Yo.I., K.H., F.T., Ta.Y., K.K., Y.N., R.K., H.W., J.N., M.D., Y.O., and G.S. were involved in sample collection and management of the discovery cohort (JaCALS). M.M., I.I., and I.Y. provided Japanese ALS samples for the replication cohort. G.T., S.M., T.K., M.H., H.F., S.N., F.O., and H.O. generated and characterized iPSCs of patients with ALS, derived motor neurons from iPSCs, and performed molecular biology studies. R.N., G.T., N.A., Y.M., M.N., and G.S. wrote the first draft of the manuscript. All authors contributed to the revision of the manuscript. M.N. and G.S. accept full responsibility for the overall content as guarantors.

## Competing interests

The authors declare the following competing interests: Y.O. is a scientific adviser of Kohjin-Bio. H.O. reports grants and personal fees from K Pharma, Inc. during the conduct of the study and personal fees from Sanbio Co., Ltd. outside the submitted work. In addition, H.O. has a patent on a therapeutic agent for amyotrophic lateral sclerosis and a composition for treatment licensed to K Pharma, Inc. All other authors declare no competing interests related to this study.

## Patient consent

Written informed consent was obtained from all study participants.

## Additional information

[1]Department of Neurology, Aichi Medical University School of Medicine, Nagakute, Japan. [2]Division of ALS Research, Aichi Medical University School of Medicine, Nagakute, Japan. [3]Public Health Informatics Unit, Department of Integrated Health Sciences, Nagoya University Graduate School of Medicine, Nagoya, Japan. [4]Keio University Regenerative Medicine Research Center, Kawasaki, Japan. [5]Division of Neurodegenerative Disease Research, Tokyo Metropolitan Institute for Geriatrics and Gerontology, Tokyo, Japan. [6]Department of Neurology, Nagoya University Graduate School of Medicine, Nagoya, Japan. [7]Department of Clinical Research Education, Nagoya University Graduate School of Medicine, Nagoya, Japan. [8]Department of Neurology, Tokushima University Graduate School of Biomedical Sciences, Tokushima, Japan. [9]Division of Neurology, Department of Internal Medicine, Jichi Medical University, Shimotsuke, Japan. [10]Department of Neurology, Faculty of Medicine and Graduate School of Medicine, Hokkaido University, Sapporo, Japan. [11]Department of Neurology, Juntendo University School of Medicine, Tokyo, Japan. [12]Department of Neurology, Toho University Faculty of Medicine, Tokyo, Japan. [13]Department of Neurology, Mie University Graduate School of Medicine, Tsu, Japan. [14]Department of Neurology, Tohoku University Graduate School of Medicine, Sendai, Japan. [15]Department of Rehabilitation Medicine, Tohoku University Graduate School of Medicine, Sendai, Japan. [16]Department of Neurology, Graduate School of Medicine, Chiba University, Chiba, Japan. [17]Department of Neurology, Vihara Hananosato Hospital, Miyoshi, Japan. [18]Department of Neurology, NHO Higashinagoya National Hospital, Nagoya, Japan. [19]Department of Neurology, Brain Research Institute, Niigata University, Niigata, Japan. [20]Advanced Treatment of Neurological Diseases Branch, Brain Research Institute, Niigata University, Niigata, Japan. [21]Department of Neurology, Okayama University Graduate School of Medicine, Dentistry and Pharmaceutical Sciences, Okayama, Japan. [22]Department of Neurology, Tokyo Metropolitan Neurological Hospital, Tokyo, Japan. [23]Department of Neurology, Gunma University Graduate School of Medicine, Maebashi, Japan. [24]Division of Neurology, NHO Sagamihara National Hospital, Sagamihara, Kanagawa, Japan. [25]Department of Neurology and Stroke Medicine, Yokohama City University Graduate School of Medicine, Yokohama, Japan. [26]Department of Neurology and Neurological Science, NucleoTIDE and PepTIDE Drug Discovery Center (TIDE), Institute of Science Tokyo, Tokyo, Japan. [27]Department of Neurology, Fukushima Medical University School of Medicine, Fukushima, Japan. [28]Department of Neurology, Graduate School of Medical Science, Kyoto Prefectural University of Medicine, Kyoto, Japan. [29]Department of Neurology, Fujita Health University, Toyoake, Japan. [30]Department of Neural iPSC Research, Institute for Medical Science of Aging, Aichi Medical University, Nagakute, Japan. [31]Aichi Medical University, Nagakute, Japan. [32]These authors contributed equally: Ryoichi Nakamura, Genki Tohnai, Naoki Atsuta, Yumi Matsuda. [33]These authors jointly supervised this work: Masahiro Nakatochi. Gen Sobue. ✉e-mail: mnakatochi@met.nagoya-u.ac.jp; sobueg@aichi-med-u.ac.jp

## On behalf of the Japanese Consortium for Amyotrophic Lateral Sclerosis research (JaCALS) study group

Ryoichi Nakamura [1,32], Genki Tohnai[2,32], Naoki Atsuta[1,32], Daisuke Ito[6], Masahisa Katsuno [6,7], Yuishin Izumi[8], Mitsuya Morita[9], Ikuko Iwata[10], Ichiro Yabe [10], Tomoko Nakazato[11], Nobutaka Hattori [11], Takehisa Hirayama[12], Osamu Kano[12], Asako Tamura[13], Naoki Suzuki [14,15], Masashi Aoki [14], Kazumoto Shibuya[16], Satoshi Kuwabara[16], Masaya Oda[17], Rina Hashimoto[18], Ikuko Aiba[18], Tomohiko Ishihara[19,20], Osamu Onodera [19], Toru Yamashita [21], Hiroyuki Ishiura[21], Kota Bokuda[22], Toshio Shimizu[22], Yoshio Ikeda[23], Kazuko Hasegawa[24], Fumiaki Tanaka[25], Takanori Yokota[26], Kazuaki Kanai[27], Yu-ichi Noto[28], Ryuji Kaji [8], Hirohisa Watanabe[29], Jun-ichi Niwa [1], Manabu Doyu[1] & Gen Sobue [2,31,33] ✉

A full list of members and their affiliations appears in the Supplementary Information.

