## [Transparent Peer Review file · Communications Biology]

A genome-wide association study identifies the *GPM6A* locus associated with age at onset in ALS

Corresponding Author: Professor Gen Sobue

This file contains all reviewer reports in order by version, followed by all author rebuttals in order by version. Parts of this Peer Review File have been redacted as indicated to maintain the confidentiality of unpublished data.

Version 0:

Reviewer comments:

Reviewer #1

(Remarks to the Author)

I only have two questions to deep in, an both around discussion of your results:

- First is about the mean AAO in japanese population (your cohort) and the cohorts of other GWAS studies in Europe or China. I think is important to remark that the older AAO in japanese population compared to the other cohorts is due to different life expectancies in these countries, and maybe it correlates both with AAO in ALS population, and maybe is that the point why the genotype markers are different in the different countries mentioned above.
- In order to follow the argumentation previously remarked in the discussion, when it is explained the results in the subpopulation with SOD1 mutations, it should be mentioned the AAO of this subcohort, and compared with the AAO of the main cohort and the replication cohort.

Reviewer #2

(Remarks to the Author)

Nakamura and colleagues report on a genome-wide association analysis on age at symptom onset in 2,015 patients with amyotrophic lateral sclerosis. They identify rs113161727 as a modifier of age at onset in ALS.

The quality control is sufficiently detailed, and QQ-plots show no considerable inflation, indicating well-behave test statistics.

While the sample size of this GWAS is modest, the authors include a replication cohort and will share the summary statistics upon publication which is an huge pro for this study.

An important remark is how the authors defined age at onset. I could not find this in the methods section. Was it age at diagnosis or age at symptom onset. If it was age at symptom onset, what were considered first symptoms? Only weakness/dysarthria, or did they include subtle changes such as fasciculations, cramps, cognitive/behavioral changes, and atypical signs such as fatigue/dyspnea or weight loss as well?

Furthermore, the authors show that they do not replicate findings from previous GWAS on age at onset in ALS. Indeed, population-specific effects can be explanation. However, for ALS susceptibility, most signals are shared between Japanese, Chinese and European ancestries, albeit not genome-wide significant. Alternatively, the risk of false positive findings due to the limited sample size for GWAS in combination with publication bias is real. I think it is at least worth mentioning that the ALS field should work towards a larger collaborative GWAS effort on modifiers of age at onset and survival with more robust results.

Minor suggestions:

The authors stratify results for SOD1-ALS. Can the authors provide a table with the specific SOD1 mutations and was there any association between rs113161727 and a specific SOD1 mutation? Can the presence of a mutation with earlier age at onset (see PMID 27261500) be included as covariate in this model?

Were the results mostly driven by SOD1-ALS? Where the results still significant after excluding SOD1-ALS patients?

Is information available for the mutation status in other ALS genes, for example FUS, TARDBP and C9orf72? As the authors

state that population specific effects are important, it is helpful to describe these population characteristics. Is it worthwhile to stratify on other genetic variants?

For visualization, maybe a Kaplan Meier curve stratified by genotype and where disease-free survival is on y-axis is more informative?

The authors investigate whether rs113161727 is an eQTL for any of the neighboring genes. They do find an eQTL effect in a set of 12 iPSC-derived motoneuron lines from ALS patients ($p = 0.0095$), which just pass correction for multiple testing (Bonferroni correction $p = 0.0475$). Given these borderline results, I think these results should be interpreted with extreme caution. Where any replicates (technical/biological) included in the analysis? Besides, I could not find this eQTL effect in a large brain-derived eQTL dataset (MetaBrain). It is worth to describe this because I think eQTL results in such a small dataset are not conclusive without validation/follow-up.

In the discussion, the authors highlight a link between GPM6A and other neuropsychiatric traits "The genomic region containing the lead SNP near GPM6A may be located at the intersection of ALS and neuropsychiatric diseases". Can the authors check if the SNP itself is associated with any of the disease of interest, for example a limited PheWAS for this SNP in the GWAS catalog?

Regarding my previous remark on sample size. Can the authors provide the sample size when referencing to the "large cohorts" in the discussion?

Line 184 please include actual p-values or other test statistics instead referring to non-replicable associations as "not substantially associated".

Reviewer #3

(Remarks to the Author)

The manuscript is well written and addresses a relevant topic for the ALS community. It is indeed relevant to assess genetic modifiers of clinical features in ALS, which the authors did thoroughly. They underpinned their finding with in vitro evidence. As it is not my level of expertise, I cannot fully guarantee that the statistical methods of the GWAS are fully appropriate. However, this reviewer recommends accept this manuscript.

However one major concern should be addressed before:

1. Have the used LCL and derived iPSCs been characterized thoroughly in the given references? A graphical presentation of the sequencing information indicating the rs113161727 SNP would be recommended for the 6 cell lines. Inclusion in the supplemental data would be sufficient.

Minor concerns:

1. Considering the high standard deviation in the qPCR results, would it be possible to add WB data to support the increased expression of GPM6A in the rs113161727 neurons?

2. Is there an GPM6A mRNA increase evident in non-neuronal cells with rs113161727?

3. please provide detailed genetic information on the analyzed SOD1 patients. Do certain specific SOD1 mutations cluster with rs113161727? Does rs113161727 impact on the AAO in patients with identical SOD1 mutations?

Version 1:

Reviewer comments:

Reviewer #2

(Remarks to the Author)

The authors answered my questions and revised the manuscript accordingly. I have no further questions.

Reviewer #3

(Remarks to the Author)

The authors provided sufficient additional information supporting their previous data. However, as also mentioned by Reviewer 2, apparently no biological/technical replicates were included for the qPCR dataset of GPM6A, which is unusual in cell culture experiments. It should at least be clearly stated that per cell line, only neuronal one differentiation was performed. Due to the high number of individual cell lines used, the result nevertheless appears plausible.

Repeated experiments/neuronal differentiations could also help to resolve the Western blot issue. It is commendable that the authors provided Western blot data. However, the quality is insufficient, as the GAPDH loading control is highly heterogeneous. They might even have missed a statistically significant finding due to the heterogeneous protein loading on their gels. It should also be recommended to try another GPM6A primary antibody as the one used showed many unspecific bands on the gel.

Version 2:

Reviewer comments:

Reviewer #3

(Remarks to the Author)

The authors addressed all the critical points that were raised before. The quality of the western Blot data is now clearly improved.

There are no further concerns from my side.

We are grateful to all the reviewers for their excellent suggestions regarding our initial submission. We found their comments extremely valuable and helpful, and have addressed each point raised by them in detail below.

Reviewer #1 (Remarks to the Author):

I only have two questions to deep in, an both around discussion of your results:

1. First is about the mean AAO in japanese population (your cohort) and the cohorts of other GWAS studies in Europe or China. I think is important to remark that the older AAO in japanese population compared to the other cohorts is due to different life expectancies in these countries, and maybe it correlates both with AAO in ALS population, and maybe is that the point why the genotype markers are different in the different countries mentioned above.

Response: We appreciate the reviewer's insightful comments and suggestions. Indeed, Japan is one of the most aged societies globally, and its demographic profile may influence the older age at onset (AAO) in Japanese patients with ALS and detection of population-specific genetic factors.

Additionally, the genetic factors affecting ALS phenotypes are known to vary significantly across ethnic groups. For example, the rs12608932 variant in *UNC13A*, which is associated with ALS incidence and survival in European populations, shows no significant association with ALS incidence or survival in Japanese and Chinese populations (Iida et al., 2011; Chen *et al.* 2014; Yang et al., 2019; Benyamin et al., 2017; Nakamura et al., 2020; Nakamura et al., 2023). Similarly, while intermediate-length polyglutamine expansions in *ATXN2* are associated with an increased risk of ALS and shorter survival period in European and Chinese populations, these findings were not replicated in Japanese patients with ALS (Naruse et al., 2019). These studies suggest that genetic modifiers of ALS phenotypes can differ significantly among ethnic groups, emphasizing the importance of population-specific studies.

Accordingly, we have expanded the discussion to include these findings and their implications, reinforcing the importance of considering demographic and genetic diversity in ALS research (Page 11, Lines 248–266). This perspective not only highlights the unique genetic architecture in Japanese patients but also underscores the value of global efforts to uncover shared and population-specific genetic factors.

References:

Benyamin, B. *et al.* Cross-ethnic meta-analysis identifies association of the GPX3-TNIP1 locus with amyotrophic lateral sclerosis. *Nat. Commun.* **8**, 611 (2017).

- Chen, X. *et al.* Association analysis of four candidate genetic variants with sporadic amyotrophic lateral sclerosis in a Chinese population. *Neurol. Sci.* **35**, 1089–1095 (2014).
- Iida, A. *et al.* Replication analysis of SNPs on 9p21.2 and 19p13.3 with amyotrophic lateral sclerosis in East Asians. *Neurobiol. Aging* **32**, 757.e13–4 (2011).
- Nakamura, R. *et al.* A multi-ethnic meta-analysis identifies novel genes, including ACSL5, associated with amyotrophic lateral sclerosis. *Commun Biol* **3**, 526 (2020).
- Nakamura, R. *et al.* Genetic factors affecting survival in Japanese patients with sporadic amyotrophic lateral sclerosis: a genome-wide association study and verification in iPSC-derived motor neurons from patients. *J. Neurol. Neurosurg. Psychiatry* **94**, 816–824 (2023).
- Naruse, H. *et al.* Association of ATXN2 intermediate-length CAG repeats with amyotrophic lateral sclerosis correlates with the distributions of normal CAG repeat alleles among individual ethnic populations. *Neurogenetics* **20**, 65–71 (2019).
- Yang, B. *et al.* UNC13A variant rs12608932 is associated with increased risk of amyotrophic lateral sclerosis and reduced patient survival: a meta-analysis. *Neurol. Sci.* **40**, 2293–2302 (2019).

Revised Manuscript:

‘However, our study did not replicate these recently reported SNPs, suggesting that the genetic factors affecting the AAO of Japanese patients with ALS are distinct from those of patients with ALS in other populations. This discrepancy underscores the importance of genetic diversity in the modifier genes that affect ALS phenotypes and highlights the potential significance of population-specific genetic factors. **Previous studies have shown that some genetic modifiers associated with ALS in European and Chinese populations are not associated with ALS in Japanese cohorts. For instance, the rs12608932 variant in *UNC13A* is associated with ALS incidence and survival in European populations^{21–24}, but not in Japanese and Chinese populations^{11,12,25}. Similarly, while intermediate repeat expansions in *ATXN2* are associated with an increased risk of ALS and shorter survival in both European and Chinese populations^{26–29}, no such associations have been found in Japanese patients with ALS³⁰.**

The failure to replicate the association between previously reported SNPs and ALS could also be explained by the difference in AAO across populations. In fact, the distribution of the AAO for ALS in Japan differs from that in other countries^{14,20}. Phenotypes, such as the AAO and prognosis of ALS, show considerable variation, even among Asian countries³¹. In our study, the mean AAO was 62.0 years, with the peak AAO in the 65–70-year age group in the Japanese patients with ALS. In contrast, the mean AAO was 54.6 years, with peak ages in the 50s and 60s in the Chinese patients with ALS²⁰, and 59.9 years, with a peak age in the 60s in the European patients with ALS¹⁴. **As one of the most aged societies with the highest life expectancy globally³²,**

Japan's demographic profile may contribute to an older AAO distribution in patients with ALS and influence the detection of genetic factors.'

2. In order to follow the argumentation previously remarked in the discussion, when it is explained the results in the subpopulation with SOD1 mutations, it should be mentioned the AAO of this subcohort, and compared with the AAO of the main cohort and the replication cohort.

Response: We appreciate the reviewer's request for further details. Accordingly, we have added the comparisons of AAO between patients with *SOD1*-ALS and those without *SOD1* mutations in the discovery and replication cohorts to the Results (Page 7, Lines 153–158) and Supplementary Table 2. Additionally, we have included details regarding the pathogenic variants in patients with *SOD1*-ALS in Supplementary Table 3.

Specifically, the mean AAO of patients with *SOD1*-ALS (55.4 ± 12.0 years) was significantly younger than that of patients without *SOD1* mutations (62.2 ± 12.4 years in the discovery cohort, $p = 1.36 \times 10^{-5}$; 64.4 ± 11.3 years in the replication cohort, $p = 8.30 \times 10^{-8}$; Student's t-test), consistent with a previous report (Opie-Martin et al., 2022). Among patients with *SOD1*-ALS, those harboring the AG or AA genotype of rs113161727 exhibited an approximately 10 years earlier AAO compared to those with the GG genotype. These findings highlight the modifying role of rs113161727 on the AAO in this subgroup of patients.

Reference:

Opie-Martin, S. *et al.* The SOD1-mediated ALS phenotype shows a decoupling between age of symptom onset and disease duration. *Nat. Commun.* **13**, 6901 (2022).

Revised Manuscript:

We included 65 patients with ALS and *SOD1* mutations (*SOD1*-ALS) in the discovery cohort. Their mean AAO (55.4 ± 12.0 years) was significantly younger than that of patients without *SOD1* mutations (62.2 ± 12.4 years in the discovery cohort, $p = 1.36 \times 10^{-5}$; 64.4 ± 11.3 years in the replication cohort, $p = 8.30 \times 10^{-8}$; Student's t-test; Supplementary Table 2). Detailed information on *SOD1* mutations is provided in Supplementary Table 3.

Reviewer #2 (Remarks to the Author):

Nakamura and colleagues report on a genome-wide association analysis on age at symptom onset in 2,015 patients with amyotrophic lateral sclerosis. They identify rs113161727 as a modifier of age at onset in ALS.

The quality control is sufficiently detailed, and QQ-plots show no considerable inflation, indicating well-behave test statistics.

While the sample size of this GWAS is modest, the authors include a replication cohort and will share the summary statistics upon publication which is an huge pro for this study.

1. An important remark is how the authors defined age at onset. I could not find this in the methods section. Was it age at diagnosis or age at symptom onset. If it was age at symptom onset, what were considered first symptoms? Only weakness/dysarthria, or did they include subtle changes such as fasciculations, cramps, cognitive/behavioral changes, and atypical signs such as fatigue/dyspnea or weight loss as well?

Response: We thank the reviewer for bringing this to our attention. We apologize for omitting the definition of age at onset in the Methods section. We have clarified this in the revised manuscript. We defined the age at onset as the age at symptom onset. Disease onset was defined as the time when patients became initially aware of muscle weakness or impairment of swallowing, speech, or respiration, consistent with our previous articles (Yokoi et al. 2016, Nakamura et al. 2013). We have added the following to the Methods (Page 15, Lines 335–337):

‘AAO was defined as the age at which the patients first noticed muscle weakness or impairment of swallowing, speech, or respiration.’^{3,50}

References:

Nakamura, R. et al. Neck weakness is a potent prognostic factor in sporadic amyotrophic lateral sclerosis patients. *J. Neurol. Neurosurg. Psychiatry* **84**, 1365–1371 (2013).

Yokoi, D. et al. Age of onset differentially influences the progression of regional dysfunction in sporadic amyotrophic lateral sclerosis. *J. Neurol.* **263**, 1129–1136 (2016).

2. Furthermore, the authors show that they do not replicate findings from previous GWAS on

age at onset in ALS. Indeed, population-specific effects can be explanation. However, for ALS susceptibility, most signals are shared between Japanese, Chinese and European ancestries, albeit not genome-wide significant. Alternatively, the risk of false positive findings due to the limited sample size for GWAS in combination with publication bias is real. I think it is at least worth mentioning that the ALS field should work towards a larger collaborative GWAS effort on modifiers of age at onset and survival with more robust results.

Response: We appreciate the reviewer for bringing up this important point. Accordingly, we considered the potential reasons for the lack of replication of previously reported SNPs in our study.

First, we hypothesized that rs113161727 at the *ADAM29-GPM6A* locus represents a population-specific factor. Genetic modifiers that affect ALS phenotypes vary significantly across ethnic groups. For instance, the rs12608932 variant in *UNC13A*, associated with ALS incidence and survival in European populations, has shown no significant association with ALS incidence or survival in Japanese and Chinese populations (Iida et al., 2011; Chen *et al.* 2014; Yang et al., 2019; Benyamin et al., 2017; Nakamura et al., 2020; Nakamura et al., 2023). Similarly, while intermediate-length polyglutamine expansions in *ATXN2* are associated with an increased risk of ALS and shorter survival in European and Chinese populations, these findings were not replicated in Japanese patients with ALS (Naruse H, et al., 2019). These findings suggest that genetic modifiers of ALS phenotypes can differ significantly among ethnic groups, emphasizing the importance of population-specific studies. We have expanded the discussion to include these findings (Page 11, Lines 248–255).

Second, Japan is one of the most aged societies, and its demographic profile may influence the older AAO distribution in Japanese patients with ALS and the detection of population-specific genetic factors.

We also acknowledge the possibility that our findings may be false positives due to our study's relatively small sample size. To address this concern, we conducted a replication analysis in an independent Japanese ALS cohort, in which we successfully replicated the association of rs113161727 at the *ADAM29-GPM6A* locus ($p = 6.81 \times 10^{-3}$, $\beta = -5.10$, $SE = 1.87$). This replication supports the validity of our findings and reduces the likelihood of rs113161727 being a false-positive result.

Accordingly, we have revised the Discussion to address the limitations of our study and the importance of future collaborative efforts (Page 14, Lines 314–319).

References:

- Benyamin, B. *et al.* Cross-ethnic meta-analysis identifies association of the GPX3-TNIP1 locus with amyotrophic lateral sclerosis. *Nat. Commun.* **8**, 611 (2017).
- Chen, X. *et al.* Association analysis of four candidate genetic variants with sporadic amyotrophic lateral sclerosis in a Chinese population. *Neurol. Sci.* **35**, 1089–1095 (2014).
- Iida, A. *et al.* Replication analysis of SNPs on 9p21.2 and 19p13.3 with amyotrophic lateral sclerosis in East Asians. *Neurobiol. Aging* **32**, 757.e13–4 (2011).
- Nakamura, R. *et al.* A multi-ethnic meta-analysis identifies novel genes, including ACSL5, associated with amyotrophic lateral sclerosis. *Commun Biol* **3**, 526 (2020).
- Nakamura, R. *et al.* Genetic factors affecting survival in Japanese patients with sporadic amyotrophic lateral sclerosis: a genome-wide association study and verification in iPSC-derived motor neurons from patients. *J. Neurol. Neurosurg. Psychiatry* **94**, 816–824 (2023).
- Naruse, H. *et al.* Association of ATXN2 intermediate-length CAG repeats with amyotrophic lateral sclerosis correlates with the distributions of normal CAG repeat alleles among individual ethnic populations. *Neurogenetics* **20**, 65–71 (2019).
- Yang, B. *et al.* UNC13A variant rs12608932 is associated with increased risk of amyotrophic lateral sclerosis and reduced patient survival: a meta-analysis. *Neurol. Sci.* **40**, 2293–2302 (2019).

Revised Manuscripts: ‘However, our study did not replicate these recently reported SNPs, suggesting that the genetic factors affecting the AAO of Japanese patients with ALS are distinct from those of patients with ALS in other populations. This discrepancy underscores the importance of genetic diversity in the modifier genes that affect ALS phenotypes and highlights the potential significance of population-specific genetic factors. **Previous studies have shown that some genetic modifiers associated with ALS in European and Chinese populations are not associated with ALS in Japanese cohorts. For instance, the rs12608932 variant in *UNC13A* is associated with ALS incidence and survival in European populations^{21–24}, but not in Japanese and Chinese populations^{11,12,25}. Similarly, while intermediate repeat expansions in *ATXN2* are associated with an increased risk of ALS and shorter survival in both European and Chinese populations^{26–29}, no such associations have been found in Japanese patients with ALS³⁰.**’

And

‘While our findings emphasize the importance of population-specific genetic factors, our study has some limitations. The relatively modest sample size may have reduced the power to detect additional associations and contributed to the lack of replication of previously reported SNPs. Addressing these limitations will require larger, collaborative studies to identify robust genetic

modifiers of ALS phenotypes such as AAO and survival.’

Supplementary Table 3. Genetic variants, age at onset, and rs113161727 genotype distribution in 65 patients with *SOD1*-ALS

Nucleotide change	Amino acid change	Number of patients	Age at onset (years, mean \pm SD)	Number of patients with rs113161727 genotype	
				AA/AG	GG
c.10A>G	p.Lys4Glu	2	50.0 \pm 3.2	1	1
c.14C>A	p.Ala5Asp	1	76.4		1
c.20G>A	p.Cys7Tyr	3	63.0 \pm 8.1		3
c.25C>G	p.Leu9Val	1	62.9		1
c.37_54dup18	p.Gly13_Ile18dup	1	51.1	1	
c.44T>C	p.Val15Ala	2	69.6 \pm 2.9	1	1
c.116T>G	p.Leu39Arg	1	55.7	1	
c.131A>G	p.His44Arg	3	60.9 \pm 15.5	1	2
c.140A>G	p.His47Arg	4	42.9 \pm 9.8	1	3
c.280G>A	p.Gly94Ser	14	52.9 \pm 9.4	3	11
c.281G>T	p.Gly94Val	1	47.8		1
c.319C>G	p.Leu107Val	4	41.9 \pm 9.1		4
c.335G>A	p.Cys112Tyr	1	54.3		1
c.341T>C	p.Ile114Thr	2	63.7 \pm 3.8		2
c.380T>C	p.Leu127Ser	14	59.5 \pm 12.8	3	11
c.404G>A	p.Ser135Asn	6	56.1 \pm 12.9	1	5
c.425G>A	p.Gly142Glu	1	67.2		1
c.425G>C	p.Gly142Ala	1	61.7		1
c.435delGinsCGTTTA	p.Leu145Phefs*3	1	40.1		1
c.437C>T	p.Ala146Val	1	60.2		1
c.449T>C	p.Ile150Thr	1	37.7		1
Total		65	55.4 \pm 12	13	52

[Redacted line of text]

[Redacted line of text]

[Redacted block of text consisting of six lines]

[Redacted block of text consisting of two lines]

[Redacted]	[Redacted]	[Redacted]	[Redacted]	[Redacted]	[Redacted]	[Redacted]	[Redacted]	[Redacted]	[Redacted]	[Redacted]
[Redacted]	[Redacted]	[Redacted]	[Redacted]	[Redacted]	[Redacted]	[Redacted]	[Redacted]	[Redacted]	[Redacted]	[Redacted]

[Redacted block of text consisting of three lines]

2. Were the results mostly driven by SOD1-ALS? Where the results still significant after excluding SOD1-ALS patients?

Response: We appreciate the reviewer for bringing up this important point. As shown in Supplementary Table 4, excluding patients with *SOD1*-ALS from the discovery cohort resulted in an effect size of -3.93 years (SE: 0.80, $p = 8.7 \times 10^{-7}$), indicating that the p-value and effect size were largely unchanged. To further validate our findings, we excluded one patient with *SOD1*-ALS from the replication cohort and performed a meta-analysis by combining the discovery and replication cohorts after removing all patients with *SOD1*-ALS. The combined analysis demonstrated genome-wide significance, with a p-value of 2.23×10^{-8} , confirming that the association between rs113161727 and AAO remains statistically robust and significant. This highlights the broader relevance of rs113161727 as an AAO modifier, beyond *SOD1*-ALS. We have added these results to the manuscript (Page 9, Lines 191–193) and Supplementary Table 5 and Supplementary Figure 4.

Revised Manuscript:

‘We also conducted a meta-analysis excluding patients with *SOD1*-ALS. The meta-analysis confirmed genome-wide significance at the 4q34.2 locus (rs113161727, $p = 2.23 \times 10^{-8}$, $\beta = -4.11$, SE = 0.73; Supplementary Figure 4, Supplementary Table 5).’

Supplementary Figure 4. Genome-wide meta-analysis for AAO in ALS patients without *SOD1* mutation

(a) Manhattan plot of the meta-analysis for AAO in Japanese patients with ALS without *SOD1* mutation.

The results from the discovery and replication cohorts excluding patients with *SOD1*-ALS were combined in a meta-analysis, which confirmed the genome-wide significance of SNPs at the 4q34.2 locus.

(b) Regional association plots for the 4q34.2 locus identified in the meta-analysis. The vertical axis represents $-\log_{10}(p\text{-value})$ for assessing the association between each SNP and AAO. Colors indicate the linkage disequilibrium (r^2) between each neighboring SNP and rs113161727 based on the JPT population of the 1000 Genomes Project Phase 3. The SNP rs113161727 is the lead SNP in the meta-analysis which includes all patients, but not in the analysis excluding patients with *SOD1*-ALS.

(c) Q-Q plot for the p-values in the meta-analysis. The vertical and horizontal axes indicate the observed and expected $-\log_{10}(p\text{-value})$ for tests of association between the SNPs and AAO in patients with ALS, respectively.

Abbreviations: SNP, single-nucleotide polymorphism; JPT, Japanese people in Tokyo, Japan; ALS, amyotrophic lateral sclerosis; AAO, age at onset.

[Redacted]

[Redacted]

[Redacted]

[Redacted]

[REDACTED]

4. For visualization, maybe a Kaplan Meier curve stratified by genotype and where disease-free survival is on y-axis is more informative?

Response: We thank the reviewer for this important comment. In designing our statistical analysis, we carefully considered the use of Cox proportional hazards models and Kaplan–Meier curves for AAO in ALS. However, we ultimately decided against these methods because our dataset contains no censored data, thus—all individuals in our study have a recorded AAO. As Cox models are primarily designed for survival data including censored cases, their application would not provide additional statistical power or meaningful insights in this context.

Instead, we used BOLT-LMM, which is well-suited for continuous traits such as AAO, while also accounting for the population structure and genetic relatedness. To complement our GWAS findings, we visualized the distribution of AAO across genotypes using box plots and histograms, which we believe offer a more intuitive and interpretable representation of the data.

Accordingly, we have included Supplementary Figure 2, which presents the Cumulative distribution of age at onset stratified by genotype. We hope that this addition sufficiently addresses the reviewer’s request.

Supplementary Figure 2. Cumulative distribution of age at onset in Japanese patients with ALS stratified by rs113161727 genotype

Cumulative distribution of age at onset in Japanese patients with ALS: (a) in the discovery cohort and (b) in patients with *SOD1*-ALS, stratified by rs113161727 genotype.

5. The authors investigate whether rs113161727 is an eQTL for any of the neighboring genes. They do find an eQTL effect in a set of 12 iPSC-derived motor neuron lines from ALS patients ($p = 0.0095$), which just pass correction for multiple testing (Bonferroni correction $p = 0.0475$). Given these borderline results, I think these results should be interpreted with extreme caution. Where any replicates (technical/biological) included in the analysis? Besides, I could not find this eQTL effect in a large brain-derived eQTL dataset (MetaBrain). It is worth to describe this because I think eQTL results in such a small dataset are not conclusive without validation/follow-up.

Response: We appreciate the reviewer's insightful comments.

In our original analysis, we used RT-qPCR to evaluate *GPM6A* expression in iPSC-derived motor neurons from 12 patients with ALS, observing a significant difference between AG and GG genotypes ($p = 0.0095$). To further substantiate these findings, we increased the sample size of the RT-qPCR analysis from 12 to 20 patients (AG: $n = 10$, GG: $n = 10$), again observing significantly elevated *GPM6A* expression in AG carriers ($p = 0.0039$, Bonferroni-adjusted $p =$

0.0195, Mann–Whitney U test). This expansion of the original dataset provides additional validation of the observed eQTL effect. Accordingly, we have updated Figure 4.

Additionally, we analyzed RNA sequencing (RNA-seq) in iPSC-derived motor neurons from 67 Japanese patients with ALS. Although these iPSCs were generated following the same method used in the RT-qPCR analysis, the RNA samples were collected and processed in a separate research laboratory using a different motor neuron differentiation protocol. Consistent with the RT-qPCR results, *GPM6A* expression levels were significantly elevated in patients with the rs113161727 AG genotype compared to those with the GG genotype ($p = 0.029$, Mann–Whitney U test). This result further supports the robustness of the observed eQTL effect.

Importantly, the observed differences between our dataset and the MetaBrain resource can be partly attributed to the contrasting tissue origins and demographic backgrounds. MetaBrain is primarily derived from adult brain tissue of predominantly European individuals, while our analyses are based on iPSC-derived motor neurons from Japanese patients with ALS. These differences in tissue context and population genetic architecture may account for the discrepancies in eQTL signals.

We recognize the limitations of the current dataset and agree that further validation in independent cohorts will be important.

Accordingly, we have added these results (Page 10, Lines 211–230), the corresponding methods (Page 22, Lines 491–524), and Supplementary Figure 6.

Revised Manuscript:

Results section

‘We investigated the effects of rs113161727 on the expression of its surrounding genes (*GLRA3*, *ADAM29*, *GPM6A*, *WDR17*, and *SPATA4*) in induced pluripotent stem cell (iPSC)-derived motor neurons of patients with ALS. Gene expression levels were assessed by real-time quantitative reverse transcription PCR (RT-qPCR). We established iPSCs from lymphoblastoid B-cell lines (LCLs) derived from 20 patients with ALS following established protocols^{16,17} Subsequently, we derived motor neurons from these iPSCs. The rs113161727 genotypes in both LCLs and iPSC-derived motor neurons were confirmed by Sanger sequencing (Supplementary Figure 5). In iPSC-derived motor neurons, the expression levels of *GPM6A* were significantly higher in the presence of the rs113161727 AG genotype ($N = 10$) than in that of the GG genotype ($N = 10$) (Figure 4; $p = 0.0039$, Mann–Whitney U test). In contrast, rs113161727 did not affect the expression of other surrounding genes (*GLRA3*, *ADAM29*, *WDR17*, and *SPATA4*). These results suggest that rs113161727 is associated with the upregulated expression of *GPM6A*.

To validate this finding, we further analyzed RNA sequencing (RNA-seq) data for iPSC-derived motor neurons from 67 Japanese patients with ALS. Consistent with our RT-qPCR results, the RNA-seq analysis confirmed that *GPM6A* expression was significantly higher in patients with the AG genotype (N = 9) than in those with the GG genotype (N = 58) ($p = 0.029$, Mann–Whitney *U* test, Supplementary Figure 6).’

Figure 4. Relative expression levels of genes surrounding rs113161727 in induced pluripotent stem cell (iPSC)-derived motor neurons from patients with amyotrophic lateral sclerosis (ALS) with each genotype of rs113161727.

The expression levels of genes (*GLRA3*, *ADAM29*, *GPM6A*, *WDR17*, and *SPATA4*) present surrounding rs113161727 in iPSC-derived motor neurons from 20 patients with ALS were examined using real-time quantitative reverse transcription-polymerase chain reaction. The expression levels of *GPM6A* mRNA were significantly higher in iPSC-derived motor neurons with the AG genotype of rs113161727 (N = 10) than in those with the GG genotype (N = 10, $p = 0.0039$). In contrast, expression levels of the other four genes (*GLRA3*, *ADAM29*, *WDR17*, and *SPATA4*) were not affected by the rs113161727 genotype. The mRNA expression levels of each genotype were compared using the Mann–Whitney *U* test. The mRNA levels of each gene were normalized to the levels of *ACTB*. Each black dot represents an individual sample. The bottom and top of the box indicate the interquartile range (25th and 75th percentiles) and the line represents the median. The whiskers under and over the box correspond to the minimum and maximum values. * $p < 0.01$.

Supplementary Figure 6. *GPM6A* expression stratified by rs113161727 genotype in iPSC-derived motor neurons from patients with ALS.

Box plot displaying *GPM6A* transcript levels (TPM) obtained from RNA sequencing of induced pluripotent stem cell (iPSC)-derived motor neurons from patients with ALS, stratified by rs113161727 genotype. Each gray dot represents an individual sample (AG: n = 9; GG: n = 58). The bottom and top of the box indicate the interquartile ranges (25th and 75th percentiles), and the line represents the median. Whiskers under and over the box correspond to a $1.5 \times$ interquartile range. *GPM6A* expression was significantly higher in the AG genotype compared to that in the GG genotype ($p = 0.029$, Mann–Whitney U test).

6. In the discussion, the authors highlight a link between GPM6A and other neuropsychiatric traits “The genomic region containing the lead SNP near GPM6A may be located at the intersection of ALS and neuropsychiatric diseases”. Can the authors check if the SNP itself is associated with any of the disease of interest, for example a limited PheWAS for this SNP in the GWAS catalog?

Response: We appreciate the reviewer’s insightful comment. We initially hypothesized that the genomic region containing rs113161727 might be located at the intersection of ALS and neuropsychiatric diseases. Following the reviewer’s suggestion, we checked for associations of rs113161727 with other diseases of interest in the GWAS Catalog database and relevant literature (McLaughlin et al, 2017; Li et al, 2021; Van Rheenen et al, 2021). No significant associations were observed. To ensure that our interpretation accurately reflects the available evidence, we

have removed the statement ‘The genomic region containing the lead SNP near *GPM6A* may be located at the intersection of ALS and neuropsychiatric diseases.’ The revised discussion now focuses on the broader role of *GPM6A* in neuropsychiatric disorders while acknowledging the absence of a direct association between rs113161727 and these traits (Page 13, Lines 282–290).

References:

- Li, C., Yang, T., Ou, R. & Shang, H. Overlapping genetic architecture between schizophrenia and neurodegenerative disorders. *Front. Cell Dev. Biol.* **9**, 797072 (2021).
- McLaughlin, R. L. *et al.* Genetic correlation between amyotrophic lateral sclerosis and schizophrenia. *Nat. Commun.* **8**, 14774 (2017).
- Van Rheenen, W. *et al.* Common and rare variant association analyses in amyotrophic lateral sclerosis identify 15 risk loci with distinct genetic architectures and neuron-specific biology. *Nat. Genet.* **53**, 1636–1648 (2021).

Revised Manuscript:

‘The lead SNP in our study, rs113161727, affected the expression of *GPM6A*. *GPM6A* encodes glycoprotein M6a (GPM6A). Several SNPs in *GPM6A* have been associated with various neuropsychiatric diseases, such as schizophrenia and depression^{25,26}. Patients with ALS and their relatives often have comorbidities such as anxiety, depression, cognitive dysfunction, and suicidal ideation^{27,28}. Recent studies have shown that ALS, neuropsychiatric diseases, and cognitive dysfunction share common genetic backgrounds^{10,29,30}. ~~The genomic region containing the lead SNP near *GPM6A* may be located at the intersection of ALS and neuropsychiatric diseases.~~ These findings suggest that *GPM6A* may play a role in shared pathways between ALS and neuropsychiatric diseases. Further studies are needed to elucidate its role and underlying mechanisms.’

7. Regarding my previous remark on sample size. Can the authors provide the sample size when referencing to the “large cohorts” in the discussion?

Response: We thank the reviewer for pointing this out. We have added the specific sample sizes to the Discussion as suggested (Page 11, Lines 241–242).

Revised Manuscript:

‘However, a recent study revealed a significant association between *CTIF* polymorphisms and AAO in a large cohort comprising 9,353 European patients with ALS¹⁴. Additionally, a GWAS in 2,788 patients with ALS of Chinese ancestry identified SNPs in *FRMD* associated with earlier AAO (by 3.15 years) for ALS²⁰.’

8. Line 184 please include actual p-values or other test statistics instead referring to non-replicable associations as “not substantially associated”.

Response: In accordance with the reviewer’s suggestion, we have modified the description of the previously reported SNPs to include the actual p-values and test statistics (Page 9, Lines 197–198).

Revised Manuscript:

‘The previously reported SNPs in European and Chinese patients with ALS, such as rs2046243 in *CTIF* and rs10128627 in *FRMD8*, showed no significant association with AAO in our study (rs2046243: $p = 0.466$, effect = -0.393, SE = 0.538; rs10128627: $p = 0.384$, effect = 0.597, SE = 0.686).’

Reviewer #3 (Remarks to the Author):

The manuscript is well written and addresses a relevant topic for the ALS community. It is indeed relevant to assess genetic modifiers of clinical features in ALS, which the authors did thoroughly. They underpinned their finding with in vitro evidence. As it is not my level of expertise, I cannot fully guarantee that the statistical methods of the GWAS are fully appropriate. However, this reviewer recommends accept this manuscript.

However one major concern should be addressed before:

1. Have the used LCL and derived iPSCs been characterized thoroughly in the given references? A graphical presentation of the sequencing information indicating the rs113161727 SNP would be recommended for the 6 cell lines. Inclusion in the supplemental data would be sufficient.

Response: We sincerely thank the reviewer for this important comment. To address this point, we confirmed the rs113161727 genotypes in both LCLs and iPSC-derived motor neurons using Sanger sequencing. To enhance the robustness of the RT-qPCR results, we expanded the analysis from 12 to 20 iPSC-derived motor neuron lines (AG: n = 10; GG: n = 10). Sanger sequencing was also performed in all 20 lines to verify the rs113161727 genotypes in both LCLs and iPSC-derived motor neurons. Sequencing chromatograms clearly showed the rs113161727 AG and GG genotypes in both cell types. These data have been included as Supplementary Figure 5 and the manuscript (Page 10, Lines 217–219 and Page 20, Lines 464–467).

Revised Manuscripts:

Results section

The rs113161727 genotypes in both LCLs and iPSCs were confirmed by Sanger sequencing (Supplementary Figure 5). Ten patients carried the AG genotype, while the other ten had the GG genotype.

Methods section

Validation of LCL and iPSC-derived motor neuron genotypes

iPSC-derived motor neurons were established from LCLs derived from 20 Japanese patients with ALS. The rs113161727 genotypes in both LCLs and iPSC-derived motor neurons were confirmed using Sanger sequencing.

a)

	LCLs	iPSC-derived motor neurons	rs113161727
sALS-001			G/G
sALS-002			G/G
sALS-003			G/G
sALS-004			G/G
sALS-005			G/G
sALS-006			G/G
sALS-007			G/G
sALS-008			G/G
sALS-009			G/G
sALS-010			G/G

b)

Supplementary Figure 5. Confirmation of rs113161727 genotypes in LCLs and iPSC-derived motor neurons by Sanger sequencing

Sequencing chromatograms showing the rs113161727 genotypes in LCLs and iPSC-derived motor neurons from patients with ALS used for RT-qPCR.

(a) Sequencing chromatograms from patients with the GG genotype confirm homozygosity at the rs113161727 locus in both cell types. (b) Sequencing chromatograms from patients with the AG genotype confirm the presence of both alleles at the rs113161727 locus in LCLs and iPSC-derived motor neurons. Abbreviations: LCLs, lymphoblastoid B-cell lines; iPSC, induced pluripotent stem cell; RT-qPCR, real-time quantitative reverse transcription-polymerase chain reaction.

Minor concerns:

1. Considering the high standard deviation in the qPCR results, would it be possible to add WB data to support the increased expression of *GPM6A* in the rs113161727 neurons?

Response:

We appreciate the reviewer's insightful comment. In our original analysis, we performed RT-qPCR on iPSC-derived motor neurons from 12 patients with ALS (AG: n = 6, GG: n = 6), which showed significantly elevated *GPM6A* expression in AG genotype carriers ($p = 0.0095$). To strengthen these findings, we increased the number of RT-qPCR samples to 20 (AG: n = 10, GG: n = 10). This expanded analysis again demonstrated significantly higher *GPM6A* expression in AG carriers ($p = 0.0039$, Mann–Whitney *U* test). Accordingly, we have updated Figure 4.

In addition, we performed RNA sequencing (RNA-seq) on iPSC-derived motor neurons from 67 Japanese patients with ALS generated using a different motor neuron differentiation protocol. Consistent with our RT-qPCR results, the RNA-seq analysis confirmed a significantly higher *GPM6A* expression in the AG genotype carriers (n = 9) compared to that in the GG genotype carriers (n = 58) ($p = 0.029$, Mann–Whitney *U* test; Supplementary Figure 6). We have revised the Results (Page 10, Lines 211–230).

We also evaluated *GPM6A* protein levels using Western blotting in a subset of 10 iPSC-derived motor neuron samples (AG: n = 5, GG: n = 5, Figure R3). Although a trend toward higher expression in AG carriers was observed, the difference was not statistically significant. This may reflect variability in protein expression levels or the limited sensitivity of the assay.

Figure 4. Relative expression levels of genes surrounding rs113161727 in induced pluripotent stem cell (iPSC)-derived motor neurons from patients with amyotrophic lateral sclerosis (ALS) with each genotype of rs113161727.

The expression levels of genes (*GLRA3*, *ADAM29*, *GPM6A*, *WDR17*, and *SPATA4*) present surrounding rs113161727 in iPSC-derived motor neurons from 20 patients with ALS were

examined using real-time quantitative reverse transcription-polymerase chain reaction. The expression levels of *GPM6A* mRNA were significantly higher in iPSC-derived motor neurons with the AG genotype of rs113161727 (N = 10) than in those with the GG genotype (N = 10, $p = 0.0039$). In contrast, expression levels of the other four genes (GLRA3, ADAM29, WDR17, and SPATA4) were not affected by the rs113161727 genotype. The mRNA expression levels of each genotype were compared using the Mann–Whitney U test. The mRNA levels of each gene were normalized to the levels of ACTB. Each black dot represents an individual sample. The bottom and top of the box indicate the interquartile range (25th and 75th percentiles) and the line represents the median. The whiskers under and over the box correspond to the minimum and maximum values. * $p < 0.01$.

Supplementary Figure 6. *GPM6A* expression stratified by rs113161727 genotype in iPSC-derived motor neurons from patients with ALS.

Box plot displaying *GPM6A* transcript levels (TPM) obtained from RNA sequencing of induced pluripotent stem cell (iPSC)-derived motor neurons from patients with ALS, stratified by rs113161727 genotype. Each gray dot represents an individual sample (AG: $n = 9$; GG: $n = 58$). The bottom and top of the box indicate the interquartile ranges (25th and 75th percentiles), and the line represents the median. Whiskers under and over the box correspond to a $1.5 \times$ interquartile range. *GPM6A* expression was significantly higher in the AG genotype compared to that in the GG genotype ($p = 0.029$, Mann–Whitney U test).

Figure R3. Western blot analysis of GPM6A protein expression in iPSC-derived motor neurons from AG and GG genotype carriers.

To evaluate the protein expression level of GPM6A, Western blot analysis was performed using samples from five patients in each genotype group (AG and GG). SH-SY5Y cells transfected with GPM6A-GFP were used as a positive control. The antibodies used were anti-GPM6A (Thermo #720252) and anti-GAPDH (Merck MAB374).

2. Is there an GPM6A mRNA increase evident in non-neuronal cells with rs113161727?

Response: We thank the reviewer for this insightful question. In our study, we primarily focused on iPSC-derived motor neurons because *GPM6A* is predominantly expressed in neuronal tissues and plays a critical role in neuronal development and synapse formation. To address the reviewer's concern regarding the cell-type specificity of the observed upregulation, we performed *GPM6A* mRNA expression analysis in lymphoblastoid cell lines (LCLs). Our additional analyses of these non-neuronal LCLs did not reveal any significant differences in *GPM6A* expression between the rs113161727 genotypes. According to the GTEx v8 portal (<https://www.gtexportal.org/home/gene/GPM6A>, Figure R4), while GPM6A is highly expressed in brain tissues, its expression in non-neuronal tissues including whole blood, is much lower. These findings suggest that the upregulation of *GPM6A* associated with rs113161727 is likely to be a neuron-specific phenomenon.

Bulk tissue gene expression for *GPM6A* (ENSG00000150625.17)

Figure R4. Bulk tissue gene expression of *GPM6A* from the GTEx v8 portal.

Supplementary Table 3. Genetic variants, age at onset, and rs113161727 genotype distribution in 65 patients with *SOD1*-ALS

Nucleotide change	Amino acid change	Number of patients	Age at onset (years, mean \pm SD)	Number of patients with rs113161727 genotype	
				AA/AG	GG
c.10A>G	p.Lys4Glu	2	50.0 \pm 3.2	1	1
c.14C>A	p.Ala5Asp	1	76.4		1
c.20G>A	p.Cys7Tyr	3	63.0 \pm 8.1		3
c.25C>G	p.Leu9Val	1	62.9		1
c.37_54dup18	p.Gly13_Ile18dup	1	51.1	1	
c.44T>C	p.Val15Ala	2	69.6 \pm 2.9	1	1
c.116T>G	p.Leu39Arg	1	55.7	1	
c.131A>G	p.His44Arg	3	60.9 \pm 15.5	1	2
c.140A>G	p.His47Arg	4	42.9 \pm 9.8	1	3
c.280G>A	p.Gly94Ser	14	52.9 \pm 9.4	3	11
c.281G>T	p.Gly94Val	1	47.8		1
c.319C>G	p.Leu107Val	4	41.9 \pm 9.1		4
c.335G>A	p.Cys112Tyr	1	54.3		1
c.341T>C	p.Ile114Thr	2	63.7 \pm 3.8		2
c.380T>C	p.Leu127Ser	14	59.5 \pm 12.8	3	11
c.404G>A	p.Ser135Asn	6	56.1 \pm 12.9	1	5
c.425G>A	p.Gly142Glu	1	67.2		1
c.425G>C	p.Gly142Ala	1	61.7		1
c.435delGinsCGTTTA	p.Leu145Phefs*3	1	40.1		1
c.437C>T	p.Ala146Val	1	60.2		1
c.449T>C	p.Ile150Thr	1	37.7		1
Total		65	55.4 \pm 12	13	52

[Redacted text block]

[Redacted text block]

[Redacted text block]

[Redacted]	[Redacted]	[Redacted]	[Redacted]	[Redacted]	[Redacted]	[Redacted]	[Redacted]	[Redacted]	[Redacted]	[Redacted]
[Redacted]	[Redacted]	[Redacted]	[Redacted]	[Redacted]	[Redacted]	[Redacted]	[Redacted]	[Redacted]	[Redacted]	[Redacted]

[Redacted text block]

We sincerely appreciate the time and effort of all reviewers in evaluating our second submission. Their comments were extremely valuable and have helped us to improve the manuscript. Our point-by-point responses to each of the reviewers' comments are provided below.

Reviewer #2 (Remarks to the Author):

The authors answered my questions and revised the manuscript accordingly. I have no further questions.

Response: We appreciate your positive feedback and your assistance in improving our manuscript. We believe that our study makes a meaningful contribution to elucidating the genetic architecture of ALS.

Reviewer #3 (Remarks to the Author):

1. The authors provided sufficient additional information supporting their previous data. However, as also mentioned by Reviewer 2, apparently no biological/technical replicates were included for the qPCR dataset of GPM6A, which is unusual in cell culture experiments. It should at least be clearly stated that per cell line, only neuronal one differentiation was performed. Due to the high number of individual cell lines used, the result nevertheless appears plausible.

Response: We appreciate the reviewer's important comment and apologize for the previous lack of clarity in our description. As correctly noted, we performed one round of motor neuron differentiation per iPSC line, followed by RT-qPCR analysis. This point has now been clearly stated in the revised Methods section of the manuscript (Pages 20–21, Lines 464–466).

“Motor neuron differentiation was performed once per iPSC line, using 20 independent patient-derived iPSC lines (AG: n = 10; GG: n = 10).”

We acknowledge the reviewer's point regarding biological replicates. Although we did not conduct biological replicates per cell line, our study involved a relatively large number (n = 20) of independently generated patient-derived iPSC lines, effectively capturing inter-individual biological variability and supporting the observed genotype-specific differences in *GPM6A* expression.

To further validate our findings, we independently performed RNA-seq analysis using iPSC-

derived motor neurons from 67 Japanese patients with ALS. This RNA-seq analysis confirmed significantly higher expression of *GPM6A* in patients carrying the AG genotype (N=9) compared to those with the GG genotype (N=58) ($p = 0.029$, Mann–Whitney *U* test; Supplementary Figure 6), providing robust support for the reproducibility of our RT-qPCR results.

Additionally, each RT-qPCR reaction was performed in technical triplicate to ensure consistency and reliability. This information has now been explicitly included in the Results and Methods section and reflected in Figure 4. Numerical source data are provided in Supplementary Data 4.

Results section (Page 10, Lines 225–226)

“RT-qPCR analyses for all genes were performed with three technical replicates per sample.”

Methods section (Pages 21–22, Lines 487–491)

“RT-qPCR analysis included technical replicates conducted in three wells for each plate. The Cq values from the three technical replicates were averaged to generate a single raw Cq value per assay. Furthermore, each experiment was independently repeated three times to ensure reproducibility. The final value represents the mean of the three independent experiments.”

We appreciate this valuable suggestion.

Figure 4. Relative expression levels of genes surrounding rs113161727 in induced pluripotent stem cell (iPSC)-derived motor neurons from patients with amyotrophic lateral sclerosis (ALS) with each genotype of rs113161727

The expression levels of genes (*GLRA3*, *ADAM29*, *GPM6A*, *WDR17*, and *SPATA4*) present surrounding rs113161727 in iPSC-derived motor neurons from 20 patients with ALS were examined using real-time quantitative reverse transcription-polymerase chain reaction. The expression levels of *GPM6A* mRNA were significantly higher in iPSC-derived motor neurons with the AG genotype of rs113161727 (N = 10) than in those with the GG genotype (N = 10, $p = 0.0039$). In contrast, expression levels of the other four genes (*GLRA3*, *ADAM29*, *WDR17*, and

SPATA4) were not affected by the rs113161727 genotype. The mRNA expression levels of each genotype were compared using the Mann–Whitney *U* test. The mRNA levels of each gene were normalized to the levels of *ACTB*. RT-qPCR analysis included technical replicates conducted in three wells for each plate. The Cq values from the three technical replicates were averaged to generate a single raw Cq value per assay. Each black dot represents an individual sample. The bottom and top of the box indicate the interquartile range (25th and 75th percentiles) and the line represents the median. The whiskers under and over the box correspond to the minimum and maximum values. * $p < 0.01$.

2. Repeated experiments/neuronal differentiations could also help to resolve the Western blot issue. It is commendable that the authors provided Western blot data. However, the quality is insufficient, as the GAPDH loading control is highly heterogeneous. They might even have missed a statistically significant finding due to the heterogeneous protein loading on their gels. It should also be recommended to try another GPM6A primary antibody as the one used showed many unspecific bands on the gel.

Response: We appreciate the reviewer's constructive feedback regarding the Western blot. To address these concerns, we carefully re-evaluated GPM6A protein expression using Western blotting on iPSC-derived motor neurons from 10 patients with ALS (rs113161727 AG genotype, $n = 5$; GG genotype, $n = 5$).

Although we initially observed heterogeneity with GAPDH as the loading control, switching to β -actin as an alternative loading control resolved this issue, leading to more consistent and reproducible results. Although higher GPM6A protein expression was observed in some AG genotype cases, the difference was not statistically significant ($p = 0.22$).

We have attached the Western blot (Figure R1) and the box plot of Western blot quantification data (Figure R2). Each Western blot experiment was independently repeated three times to ensure reproducibility. While this Western blot result provides supportive data, the lack of statistical significance indicates a need for further studies, potentially with larger sample sizes or alternative methodologies, to confirm protein-level differences.

A

B

C

Figure R1. Western blot analysis of GPM6A protein expression in iPSC-derived motor neurons from AG and GG genotype carriers

To evaluate the protein expression level of *GPM6A*, Western blot analysis was performed using

samples from five patients in each genotype group (AG and GG). SH-SY5Y cells transfected with GPM6A-GFP were used as a positive control. The antibodies used were anti-GPM6A (Thermo Fisher Scientific #720252) and anti- β -actin (Cell Signaling Technology #4970). Each Western blot experiment was independently repeated three times to ensure reproducibility.

Figure R2. GPM6A protein expression in iPSC-derived motor neurons from AG and GG genotype carriers

(A) Box plot showing GPM6A protein expression in iPSC-derived motor neurons from five patients in each genotype group (AG and GG). (B) Magnified view of Figure R2 (A).

β -actin was used as the loading control. The primary antibody for GPM6A was anti-GPM6A (Thermo Fisher Scientific #720252), and the loading control antibody was anti- β -actin (Cell Signaling Technology #4970). Band intensities were quantified using ATTO Densitograph software (ATTO, Tokyo, Japan), normalized to β -actin, and expressed relative to the mean value of the GG group, which was set to 1. Comparison between genotype groups was conducted using the Mann–Whitney U test. Although higher GPM6A protein expression was observed in some AG genotype cases, the difference was not statistically significant ($p = 0.22$). The bottom and top of the box indicate the interquartile range (25th and 75th percentiles) and the line represents the median. The whiskers under and over the box correspond to the minimum and maximum values.

We sincerely appreciate all reviewers for their time and constructive feedback, which helped in substantially improving the manuscript. Our point-by-point responses are provided below.

Reviewer #3 (Remarks to the Author):

1. The authors addressed all the critical points that were raised before. The quality of the western Blot data is now clearly improved.

There are no further concerns from my side.

Response: We thank you for your positive evaluation. We are pleased to learn that the revised western blot data address the previously raised concerns.